# Benefits of assimilating thin sea ice thickness from SMOS into the TOPAZ system

**Jiping Xie[1], Francois Counillon[1], Laurent Bertino[1], Xiangshan Tian-Kunze[2], and Lars Kaleschke[2]**

1. Nansen Environmental and Remote Sensing Center, Bergen, Norway
2. Institute of Oceanography, University of Hamburg, German

**Abstract** An observation product for thin sea ice thickness (SMOS-Ice) is
derived from the brightness temperature data of the European Space
Agency's (ESA) Soil Moisture and Ocean Salinity (SMOS) Mission. This
product is available in near-real time, at daily frequency, during the cold
season. In this study, we investigate the benefit of assimilating SMOS-Ice
into the TOPAZ coupled ocean and sea ice forecasting system, which is
the Arctic component of the Copernicus marine environment monitoring
services. The TOPAZ system assimilates sea surface temperature (SST),
altimetry data, temperature and salinity profiles, ice concentration, and ice
drift with the Ensemble Kalman Filter (EnKF). The conditions for
assimilation of sea ice thickness thinner than 0.4 m are favorable, as
observations are reliable below this threshold and their probability
distribution is comparable to that of the model. Two parallel Observing
System Experiments (OSE) have been performed in March and
November 2014, in which the thicknesses from SMOS-Ice (thinner than
0.4 m) are assimilated in addition to the standard observational data sets.
It is found that the Root Mean Square Difference (RMSD) of thin sea ice
thickness is reduced by 11% in March and 22% in November compared
to the daily thin ice thicknesses of SMOS-Ice, which suggests that
SMOS-Ice has a larger impact during the beginning of the cold season.
Validation against independent observations of ice thickness from buoys
and ice draft from moorings indicate that there are no degradations in the
pack ice but some improvements near the ice edge close to where the
SMOS-Ice has been assimilated. Assimilation of SMOS-Ice yields a slight
improvement for ice concentration and degrades neither SST nor sea
level anomaly. Analysis of the Degrees of Freedom for Signal (DFS)
indicates that the SMOS-Ice has a comparatively small impact but it has a
significant contribution in constraining the system (> 20% of the impact of
all ice and ocean observations) near the ice edge. The areas of largest
impact are the Kara Sea, the Canadian archipelago, the Baffin Bay, the
Beaufort Sea and the Greenland Sea. This study suggests that the
SMOS-Ice is a good complementary data set that can be safely included
in the TOPAZ system.

1 **Keywords**: Arctic forecasting; TOPAZ; thin sea ice thickness; SMOS-Ice;

2 Degrees of Freedom for Signal; Strongly coupled data assimilation;

# 1.    Introduction

The Arctic climate system has undergone large changes during the last 20 years: increase of temperature (Chapman and Walsh, 1993; Serreze et al., 2000; Karl et al., 2015; Roemmich et al., 2015), decrease of sea ice extent (Johannessen et al., 1999; Comiso et al., 2008; Stroeve et al., 2012), sea ice thinning and loss of sea ice volume (Rothrock et al., 1999; Kwok and Rothrock, 2009; Laxon et al., 2013). The interpretation of such changes is severely hampered by the sparseness and the complexity of the observational network. A reanalysis database can combine the sparse observations with a dynamically consistent model and is becoming an important tool.

While observations of sea ice concentrations (SIC) have been available for the past 30 years, observations of sea ice thickness (SIT) are comparatively sparse. An improved knowledge of SIT would be greatly beneficial, both for model developments and for improving the accuracy of operational ocean forecasting system. The initialization of SIT is also expected to improve predictability on seasonal time scale (Guemas et al. 2014). Until the last decade, observations of SIT were mostly limited to field campaigns or submarine measurements. Major efforts in remote sensing have been proposed to monitor the spatiotemporal evolution of SIT, and gradually obtained various products from different satellite retrieval algorithms. Measurements of thick sea ice freeboard on basin-wide scales have been derived from laser altimeters on board ICESat (e.g., Forsberg and Skourup, 2005; Kurtz et al., 2009; Kwok and Rothrock, 2009) or from radar altimeters on ERS, EnviSAT and CryoSat-2 (e.g., Laxon et al., 2003; Giles et al., 2007; Connor et al., 2009). Still, large uncertainties remain in the accuracy of the resulting SIT estimates (larger than 0.5 m) due to uncertainties in the snow depth and the sea ice density (Zygmuntowska et al., 2014). A new database based on CryoSat-2 has been provided (Laxon, 2013; Ricker et al., 2014) and has been made available in near real time (Tilling et al. 2016). Finally, methods for SIT retrieval based on measurements of the brightness temperature at a low microwave frequency of 1.4 GHz (L-band: wavelength $\lambda_a$=21 cm) have been developed in preparation for the European Space Agency's

(ESA) Soil Moisture and Ocean Salinity (SMOS) mission (Heygster et al.,
2009; Kaleschke et al., 2010; Kaleschke et al., 2013). It has been shown
that SMOS can be used to retrieve level SIT up to half a meter under cold
conditions (Kaleschke et al., 2012; Huntemann el al., 2014).
An improved retrieval method based on a radiative transfer model and a
thermodynamic sea ice model has been further proposed by considering
the variations of ice temperature, salinity and a statistical SIT distribution
(Tian-Kunze et al., 2014). An operational product has been derived from
this method and is available at daily frequency (hereafter referred to as
SMOS-Ice). The SMOS-Ice has been validated during a field campaign in
the Barents Sea (Kaleschke et al., 2016; Mecklenburg et al., 2016). It
provides daily estimate of SIT and is available since October 2010 (Tian-
Kunze et al., 2014). In this study, we are testing the benefits of
assimilating SMOS-Ice into the TOPAZ system.
The TOPAZ forecasting system (Sakov et al., 2012) is a coupled ocean-
sea ice data assimilation system and is the main Arctic Marine
Forecasting system in the Copernicus Marine Services
(http://marine.copernicus.eu/). It provides a 10-days coupled physical-
biogeochemical forecast every day and a long-term reanalysis from 1990-
2015 (Sakov et al., 2012; Xie et al., 2016). At present, TOPAZ assimilates
several data types jointly with the Ensemble Kalman Filter (EnKF): Sea
Surface Temperature (SST), along-track Sea Level Anomalies (SLA) from
satellite altimeters, in situ temperature and salinity profiles, Sea Ice
Concentration (SIC) and sea ice drift from satellites. The reanalysis
product of the TOPAZ system has been widely used in studies about
ocean circulation and sea ice in the North Atlantic Ocean or in the Arctic
region (Melsom et al., 2012; Johannessen et al., 2014; Korosov et al.,
2015; Lien et al., 2016). Although the capability for assimilating SIT has
been demonstrated in Lisæter et al. (2007), TOPAZ does not yet
assimilate SIT nor apply post-processing for this variable. The reanalysis
in the period 1991-2013 has been compared to available observations at
different periods of time (Xie et al., 2016). It was found that TOPAZ
underestimates the sea ice draft compared to in situ drafts from Sonar of

the US Navy Submarines for the period 1993-2005 (Lindsay, 2013). In spring and autumn of 2003-2008, the SITs in TOPAZ are in good agreement with those of ICESat data (Kwok and Rothrok, 2009) with a spatial correlation 0.74 and 0.84 respectively. However, the SIT in TOPAZ is too large (by more than 0.2 m) in the Beaufort Sea and too low in the rest of the Arctic (up to 1 m). When compared against the IceBridge SIT (Kurtz et al., 2013) for the period 2009-2011, it was found that the thick SIT in the central Arctic is underestimated by 1.1 m in TOPAZ. Such inaccuracies in the SIT are a common limitation of coupled ice-ocean models in the Arctic (Johnson et al., 2012; Schweiger et al., 2012; Smith et al., 2015).

The first demonstration of assimilating SMOS-Ice has been presented by Yang et al. (2014) for the period from November 2011 to January 2012. The system assimilates both SIT (thinner than 1 meter) from SMOS-Ice and SIC from Special Sensor Microwave Imager/Sounder (SSMIS) in a nested Arctic configuration of the Massachusetts Institute of Technology general circulation model (MITgcm). It uses the Localized Singular Evolutive Interpolated Kalman (LSEIK; Nerger et al., 2005) data assimilation method with a 15 members ensemble. It was found that assimilation of SMOS-Ice leads to improvement of the SIT forecasts and to a small improvement for sea ice concentration. A comparison of SIT from three moorings from the Beaufort Gyre Experiment Program (BGEP) and from one autonomous ice mass balance (IMB) buoy, shows that the overestimation of SIT is reduced. The present study follows up the work from Yang et al. (2014) but it further explores the impact and relative importance of SMOS-Ice in the perspective of an ice-ocean forecasting system: 1) the impact of assimilating SMOS-Ice is tested both during the onsets of the melting and freezing seasons; 2) SMOS-Ice is tested together with a more complete observations network and its relative contribution is quantified; 3) the results are tested with a different model at slightly higher resolution, with a comparable assimilation method but with a larger ensemble size.

This paper is organized as follows: section 2 introduces the main components of the TOPAZ system including the model, the assimilation scheme, and the observations assimilated. In section 3, we compare SMOS-Ice data to the TOPAZ reanalysis for the period 2010-2014, and investigate potential biases and whether conditions are favorable for data assimilation. In section 4, two Observing System Experiment (OSE) runs are conducted, consisting of two assimilation runs with and without the SMOS-Ice data during 2014. In Section 5, we compared the contributions of SMOS-Ice relative to other types of observations for controlling the degree of freedom of the system during assimilation.

## 2. Descriptions of the TOPAZ data assimilation system
### 2.1 The coupled ocean and sea ice model

The ocean general circulation model used in the TOPAZ system is the version 2.2 of the Hybrid Coordinate Ocean Model (HYCOM) developed at University of Miami (Bleck, 2002; Chassignet et al., 2003). HYCOM uses hybrid coordinates in the vertical, which smoothly shift from isopycnal layers in the stratified open ocean to z-level coordinates in the unstratified surface mixed layer. This feature has been demonstrated in a wide range of applications from the deep oceans to the shelf (Chassignet et al., 2009). The NERSC-HYCOM model is coupled to a one-thickness category sea ice model, for which the ice thermodynamics are described in Drange and Simonsen (1996) and the ice dynamics are based on the elastic-viscous-plastic rheology described in Hunke and Dukowicz (1997) with a modification from Bouillon et al. (2013). In the model, there is a minimum thickness of 0.1 m for both new ice and melting ice. The model grid is produced using conformal mapping (Bentsen et al., 1999) and has a quasi-homogeneous horizontal resolution of 12-16 km in the Arctic as shown in Fig. 1.

The temperatures and salinities at the model lateral boundaries are relaxed to a combined climatology of the World Ocean Atlas of 2005 (WOA05, Locarnini et al., 2006) and the version 3.0 of the Polar Science Center Hydrographic Climatology (PHC, Steele et al., 2001). A seasonal inflow is imposed at the Bering Strait with a transport that is following the

observed estimate from Woodgate et al. (2012). A balanced outflow of
similar mean transport is imposed at the southern boundary of the model.
The TOPAZ system uses atmospheric forcing from ERA-Interim (Dee et
al., 2011).
**2.2    The EnKF data assimilation**
The analysis with the standard EnKF, is expressed as follows:
$$\mathbf{X}^a = \mathbf{X}^f + \mathbf{K}(\mathbf{Y} - \mathbf{H}\mathbf{X}^f), \tag{1}.$$
where **x** is the ensemble of model state vector, the superscripts "a" and
"f" refer to the analysis and the forecast respectively. The ensemble
consists of 100 dynamical members. **H** is the observation operator and **Y**
is the perturbed observation matrix. The term innovation refers to the
misfits between the observations and the model:  i.e. the term in brackets
in equation (1). The Kalman gain **K** in Equation (1) is calculated as:
$$\mathbf{K} = \mathbf{P}^f\mathbf{H}^T[\mathbf{H}\mathbf{P}^f\mathbf{H}^T + \mathbf{R}]^{-1} \tag{2},$$
where **R** is the matrix of observation error variance and **P$^f$** is the matrix of
background error covariance, which can be calculated by an ensemble
anomalies with $N$ members - **P**= (1/$N$-1)*$\mathbf{AA}^T$. The superscript T denotes
a matrix transpose, and **A** is the ensemble of anomalies. The ensemble
anomalies is calculated as:
$\mathbf{A} = \mathbf{X} - \bar{\mathbf{x}}\mathbf{I}_N,$
where $\bar{\mathbf{x}}$ is the ensemble mean vector, and $\mathbf{I}_N = [1, ... ,1]$ is the vector with
all components equal to 1.
The TOPAZ system uses the deterministic EnKF (DEnKF, Sakov and
Oke, 2008), which is a square-root filter implementation of the EnKF that
solves the analysis without the need for perturbation of the observations.
The DEnKF overestimates the analysed error covariance by adding a
semi-definite positive term to the theoretical error covariance given by the
Kalman filter, which mitigates the need for inflation (Sakov and Oke,
31   2008).
In the DEnKF, the ensemble mean is updated by assimilating the
unperturbed observation **y**:
$\overline{\mathbf{x}^a} = \overline{\mathbf{x}}^f + \mathbf{K}(\mathbf{y} - \mathbf{H}\overline{\mathbf{x}}^f).$
The analyzed ensemble anomaly is calculated as follows:
$$\mathbf{A^a} = \mathbf{A^f} - \frac{1}{2}\mathbf{KHA^f}.$$
The full ensemble is reconstructed by adding the two terms as follows:
$$\mathbf{X^a} = \mathbf{A^a} + \overline{\mathbf{x^a}}\mathbf{I}_N \qquad (3),$$
where $\mathbf{X}^a$ is the matrix of the updated model states after assimilation.
An overview of the observations assimilated in the present TOPAZ
system is given in Table 1. Observations are quality-controlled and
superobed (i.e. the process of combining observations falling within the
same model grid cell) as in Sakov et al. (2012). TOPAZ assimilates the
following data sets on a weekly basis: the gridded SST from the
Operational Sea Surface Temperature and Sea Ice Analysis system
(OSTIA, Donlon et al., 2012); sea ice concentration from the Ocean &
Sea Ice Satellite Application Facility (OSISAF); along-track Sea Level
Anomaly by Collecte Localisation Satellites (CLS); delayed-mode profiles
of temperature and salinity from Ifremer, and the sea ice drift during the 3
days prior to the analysis from the CERSAT (Centre ERS d'Archivage et
de Traitement) of IFREMER (French Research Institute for Exploitation of
the Sea). All these standard measurements are retrieved from
http://marine.copernicus.eu. The SLA data and the sea ice drift data are
assimilated asynchronously (see Sakov et al., 2010).
**3.    Bias analyses for thin ice thickness**
The TOPAZ system has computed a reanalysis at daily frequency for
ocean and sea ice variables. Its sea ice thickness has been validated
against in situ data and satellite observations in Xie et al. (2016). Data
assimilation assumes that the model and observations errors are
unbiased. In this section, we investigate the bias by analyzing the
thickness misfits for thin sea ice during five cold seasons from 2010 to

29   2014.

SMOS-Ice products (version 2.1) are available during the cold season
(from 15[th] October to 15[th] April) at daily frequency from 2010 and up to
near-real time. The data set is provided by University of Hamburg
(Kaleschke et al., 2012; Kaleschke et al., 2013;
https://icdc.zmaw.de/1/daten/cryosphere/l3c-smos-sit.html).
Here, the daily averaged SITs of TOPAZ are compared to the
observations. The spatial or temporal bias and Root Mean Square
Difference (RMSD) are calculated as follows:

$$\mathbf{Bias} = \frac{1}{n}\sum_{i=1}^{n}(\mathbf{H}\bar{\mathbf{x}}_i^f - \mathbf{y}_i) \tag{4}$$

$$\mathbf{RMSD} = \sqrt{\frac{1}{n}\sum_{i=1}^{n}(\mathbf{H}\bar{\mathbf{x}}_i^f - \mathbf{y}_i)^2} \, , \tag{5}$$

where $\bar{\mathbf{x}}_i^f$ is compared to observations at similar time, $\mathbf{H}$ is the observation
operator (see eq. 1), and $n$ is the number of available observations within
the calculation period. Note that, we compare the TOPAZ SITs to
imperfect observations, which contains error and may also be biased. As
such, the bias as formulated in Eq. 4 refers to the difference between the
model and observation biases calculated against an unknown truth. Still it
is reasonable to assume that the bias in the observation is smaller than in
the model and that the bias obtained with Eq.4 mainly accounts for model
bias.
Figure 2 shows the simulated SIT from the TOPAZ reanalysis as
conditional expectations with respect to SMOS-Ice data sorted into bins
of 5 cm. Again, the SITs from TOPAZ in Fig.2 are selected at the same
locations and time as observations. Overall, the SIT in TOPAZ tends to
be overestimated. The overestimation varies from month to month and
with the amplitude of SIT (more pronounced for thick ice).  For SIT lower
than 0.4 m, the match between the observations and TOPAZ is relatively
good through the cold season. There is no clear bias between October
and December but a slight increasing thick bias from January-April. For
SIT larger than 0.4 m, TOPAZ clearly overestimates SIT compared to
observations during October and February-April, while it underestimates it
in November. The penetration depth for the L-Band microwave frequency
into sea ice is about 0.5 m (Kaleschke et al., 2010; Huntemann et al.,
2014), and the effect of ice melting leads to saturation beyond 0.4 m (see
Heygster et al. 2009). For these reasons, assimilation of SITs thicker than
0.4 m appears as problematic because the large bias from observations

or models may be transferred to other variables (e.g. in the ocean) via the multivariate properties of our data assimilation method (note that TOPAZ uses strongly coupled data assimilation between the ocean and sea-ice). In the following we will only assimilate the SIT observations less than 0.4 m.

We now investigate whether there is an interannual, seasonal and spatial variability in the bias of SIT. Figure 3 shows the yearly bias (as defined in Eq. 4) for SIT thinner than 0.4 m during the period 2010-2014. After 2011, the thick bias is increasing, reaching a maximum of 0.1 m in 2014. There is some seasonality in the bias, and the thick bias is larger in March than in November. There is a large spatial variability in the distribution of the bias (right panel of Fig. 3), with the bias being largest in the Beaufort Sea and in the Kara Sea. We therefore select the periods of March and November 2014 to set the assimilation system in the most difficult situations.

## 4.    Observing System Experiment of SMOS-Ice
### 4.1    Design of OSE runs for SMOS-Ice

The SMOS-Ice ice thickness data is gridded at a resolution of approximately 12.5 km and is available at daily frequency during the cold season. For the reasons explained in previous section, we only used the observations with thickness lower than 0.4 m and with a distance of at least 30 km away from the coast (See Section 3). The related innovations in Equation (1) are expressed as sea ice volume:

$$\Delta \mathbf{SIT} = \mathbf{y}_{\mathrm{smos}} - \mathbf{H}(\bar{\mathbf{h}}_{\mathrm{mod}} \times \bar{\mathbf{f}}_{\mathrm{mod}}), \qquad (6)$$

where $\mathbf{y}_{\mathrm{smos}}$ is the observed SIT for thin ice from SMOS, $\mathbf{H}$ is the same observation operator as in equation (1), $\bar{\mathbf{h}}_{\mathrm{mod}}$ is the ensemble mean of ice thickness within the grid cell and $\bar{\mathbf{f}}_{\mathrm{mod}}$ is the ensemble mean of SIC. Note that the model has a minimum thickness of 0.1 m, but SIT observations of ice thinner than 10 cm can be assimilated quantitatively because the ensemble mean from 100 ensemble members can take values as low as 1 mm. To highlight the additional impact of SMOS-Ice observations, two OSE runs are carried out:

- The **Official Run**: uses the standard observational network of the TOPAZ system. It assimilates every week the along-track Sea Level Anomaly, SST, in situ profiles of temperature and salinity, sea ice concentrations and sea ice drift data (listed in **Table** 1).

- The **Test Run**: assimilates the SMOS-Ice data in addition to the observations assimilated in the Official Run. In this study, the observation errors are assumed to be spatially uncorrelated. The observation error variance (diagonal term of **R** term in Eq. 2) for SIT is set as recommended by the provider. It is estimated based on a priori estimate of the maximum uncertainty of different input parameters: surface air temperature, bulk ice temperature and bulk ice salinity (Tian-Kunze et al., 2014). We consider an observation error variance of 25 m$^2$ to be the threshold beyond which observations are assumed fully saturated and are not assimilated in our system, this is however generally not occurring for SIT values lower than 40 cm (see Fig. 4).

Figure 4 shows the uncertainties of the observations as function of the observed thickness from SMOS in March and November of 2014. There is a linear increase of the observation error with SMOS-Ice SIT with a slope of approximately 2.6. There is no visible seasonal variation in this relation (not shown).

In the following, the two parallel OSE runs are carried out at two typical time periods of the cold season: at the onsets of the ice melting from 15$^{th}$ February to 31$^{st}$ March and at the freezing time from 15$^{th}$ October to 30$^{th}$ November in 2014.

## 4.2 Validation against assimilated measurements

The error analysis focuses on the following target quantities: SIT, SIC, SST and SLA. All quantities are derived from the ensemble mean daily averages that are compared to observations at same locations and time. The bias is calculated as specified in Eq. 4 and the RMSD as in Eq. 5.

The spatial distribution of selected SMOS-Ice data for thin sea ice is shown in the top panels of Fig. 5 during March and November of 2014. In March, the available observations in the Beaufort Sea are very few, and unevenly distributed - mainly located in the coastal areas. Hence, most of

the observations are unreliable (close to the error saturation threshold at 5 m) or too thick (> 0.4 m) to be assimilated. Therefore in the following, the results for the Beaufort Sea are only presented for November. In the middle panels of Fig. 5, the differences of RMSD for sea ice thickness between the Official Run and the Test Run are shown (red color indicates an improvement due to assimilation of SMOS-Ice and blue a degradation). In March, the improvements are mainly found to the east of Franz Josef Land and to some extent near the ice edge in the Greenland Sea. In November, the reduction of RMSD is larger than 0.2 m in the Beaufort Sea, the Greenland Sea and to the North of Svalbard. Finally, the differences of monthly ice thickness between the Official Run and the Test Run are shown in the bottom panels of Fig. 5. They suggest that assimilating SMOS-Ice leads to a reduction of sea ice thickness both in March and November 2014.

Based on Eqs. (4) and (5), the time series of daily bias and RMSD for thin ice thicknesses in the OSE runs are shown in the top panels of Fig. 6. The bias of thin SIT is reduced from 16 cm to 12 cm in March, and from 7 cm to 4 cm in November, when SMOS-Ice data is assimilated. The RMSD of thin SIT is reduced from 35 cm to 31 cm in March, and from 27 cm to 21 cm in November. This corresponds to a reduction of the bias of 25% in March and 43% in November, and a reduction of the RMSD of about 11% in March and 22% in November. In the other panels of Fig. 6, the bias and RMSD of SIC, SST and SLA are presented. There is a slight benefit for the bias and RMSD of SIC (i.e. the reduction of the SIC RMSD is about 0.001), but the statistics for SST and SLA are unchanged.

The averaged thicknesses of thin sea ice in the marginal seas - in the Kara Sea, Barents Sea and Beaufort Sea - are shown with marked lines in the panels of Fig. 7. The corresponding daily RMSDs of ice thickness relative to thin SMOS-Ice data are added with shading. In each month, there are four assimilation steps marked with vertical lines.

In the Kara Sea, the thickness observed in March is very stable with a slight gradual increase. There is a relatively uniform reduction of RMSD by about 21%, which is mainly the result from a correction of the large (too thick) bias in the model. In November, the bias is much smaller and

the resulting improvement is small (8%), but the performances are slightly improving throughout the month for RMSD.

In the Barents Sea, the observations of SIT in March show an increasing trend. The Official Run shows initially a large (thick) bias that reduces as SIT increases in the observations. Assimilation of SMOS-Ice data reduces well the initial bias, but the bias converges towards the Official Run at the end of the month and so is the RMSD. On average, the RMSD of SIT is decreased by approximately 27% from the Test Run. In November, the observations show large variability that is well captured in the Official Run but the ice is initially too thick. The RMSD reduction of the Test Run compared to the Official Run is about 19% and both the bias and the RMSD are reduced.

In the Beaufort Sea, there are too few observations to provide a representative estimate of the system performance in March (top panels of Fig. 5) and the statistics are not presented. In November, the observations show an increasing trend and the Official Run shows once again a relatively large thick bias initially. The RMSD in the Test Run is reduced by about 51%, which is mainly caused by a reduction of the bias. The increasing trend in the Test Run is in relatively good agreement with the observations.

## 4.3  Validation against independent observations of SIT and sea ice draft

Three Ice Mass Balance (IMB) buoys (Perovich et al., 2009; http://imb.erdc.dren.mil/buoyinst.htm) are available for independent validation during our period of study (*2013F, 2013G* and *2014F)*. Their drift trajectories are shown in Fig. 5 for March and November 2014. On the 1st March 2014, the buoys of *2013F* and *2013G* are located at (150.8°W, 74.8°N) and (157.9°W, 75.3°N). And on the 1st November 2014, the buoys *2013F* and *2014F* are located at (158.4°W, 77.6°N) and (146.3°W, 76.7°N) respectively. In Fig. 8, the daily SIT of the OSE runs are compared to those of the buoys along their trajectories. Between the 15th February and the 30th March, the SITs of the two runs are identical

and are increasing from 1.6 m to 1.9 m while the observations show a more moderate increase from 1.5 to 1.65 m. It should be noted that the increase in the model is not necessarily caused by thermodynamic growth only since the modeled ice motions may differ from the buoys trajectories. Between the $15^{th}$ October and the $30^{th}$ November (Buoys *2013F* and *2014F)*, the SIT in the Test Run is slightly improved compared to the Official Run (with an improvement of 2 cm). It is expected that the impact of SMOS-Ice on the two buoys is small because they are located far away from the locations where SMOS-Ice data are assimilated (shown in the top panels of Fig. 5). The TOPAZ system uses localization, meaning that the impact of observations during assimilation is limited to a certain radius and their influence reduces as function of distance. In the TOPAZ system, the effective localization radius is 90 km. Still, it is encouraging to see that the improvements seem to be increasing with time suggesting that the region influenced by SMOS-Ice is gradually spreading across the domain.

Observations of sea ice drafts from moored sonar data are another source of independent observations. There are in total 6 moorings: 2013a, 2013b, and 2013d in March 2014; 2014a, 2014b, and 2014d in November 2014, which locations are shown in Fig. 5. These measurements are available from BGEP (Kishfield et al., 2014; http://www.whoi.edu/page.do?pid=66559). They use moored upward-looking sonar instruments and collect year-round time series measurements of the sea ice draft distribution (into 0.1 m bins) at daily frequency. This data is processed to filter out wave action in the summer months that may lead to the removal of thin draft measurements (Krishfield et al., 2014). This can be problematic if the model estimates are lower than the observed values. The sea ice draft from TOPAZ is diagnosed as proposed in Alexandrov et al. (2010), i.e.:

$$d_i = h_i \frac{\rho_i}{\rho_w} + h_{sn} \frac{\rho_{sn}}{\rho_w}$$ ,

where $d_i$ is sea ice draft, $h_i$ is ice thickness, and $h_{sn}$ is the modeled snow depths. The constant $\rho_i$, $\rho_w$, and $\rho_{sn}$ are the densities for ice, water, and snow (respectively 900 kg m$^{-3}$, 1000 kg m$^{-3}$, and 300 kg m$^{-3}$). In March

2014, the observed sea ice drafts are mostly distributed between 0.8 m and 1.6 m (see Fig. 8). Both OSE runs overestimate the sea ice drafts in March, and perform identically. In November 2014, the observed sea ice drafts are thinner (< 1 m). The sea ice drafts from the OSE runs are again overestimated in all three locations. The averaged draft difference in the two runs is about 1 cm at the two moorings 2014a and 2014b, and about 16 cm at the mooring 2014d that is located closest to locations where SMOS-Ice has been assimilated (see Fig.5). We have also compared the two OSE runs in March 2014 with the NASA IceBridge SIT Quick Look data set (QL) available from National Snow and Ice Data Center. The analysis leads to similar conclusions (not shown), which is that assimilation of SMOS-Ice only yields to improvements of SIT near the ice edge near the location where SMOS-Ice is assimilated but do not yield degradation in other places.

## 5.    Relative impact of the SIT from SMOS-Ice

In this Section, the quantitative benefit of assimilating SMOS-Ice into the TOPAZ system is compared to other observations assimilated. To do so, we evaluate a performance metric calculated during the analysis, the Degree of Freedom for Signal (DFS), which is widely used for such purposes (Rodgers 2000; Cardinali et al. 2004). During the assimilation, one can calculate the DFS as follows:

$$\text{DFS} = tr\left(\frac{\partial \hat{\mathbf{y}}}{\partial \mathbf{y}}\right) = tr\left\{\frac{\partial [\mathbf{H}(\mathbf{X}^a)]}{\partial \mathbf{y}}\right\} = tr(\mathbf{KH}) \qquad (7).$$

Here, the matrix **H** is the observation operator as in equation (1), and *tr* defines the trace, applied to the matrix (**KH**). The DFS measures the reduction of mode that can be attributed to each observation type. A value of DFS close to 0 means that the observation has no impact, while a value of *m* means that the assimilation has reduced the number of degree of freedom of the ensemble by *m*. Note that the reduction cannot exceed the ensemble size; i.e. 100 here. In Sakov et al. (2012), it was recommended that the DFS should not exceed 10 % of the ensemble size to avoid a collapse of the ensemble spread.

In the following the term $\text{DFS}_{ij}$ denotes the DFS of the assimilation at time
$i$, of the $j^{th}$ type of observations, as calculated by equation (7). The
averaged DFS over a specific time period is calculated as follows:
$$\overline{\text{DFS}}_j = \frac{1}{m}\sum_{i=1}^{m}\text{DFS}_{ij}, \qquad (8).$$
where the subscript $j$ represents the $j^{th}$ type of the assimilated
observations, the subscript $i$ is time and $m$ is the total number of
assimilation steps within the considered time period (e.g. 4 for a monthly
estimate with weekly assimilation). The DFS values are calculated at
each model grid cell. In Fig. 10, we are plotting the averaged DFS maps
(as defined in Eq. 8) for the different observation data sets assimilated in
March and November. In the Arctic the total DFS is dominated by the ice
concentration that reaches large value (approximately 6) near the ice
edge. The DFS for SMOS-Ice is comparatively small and is larger in
March than in November. In some regions, the monthly DFS of SMOS-Ice
reaches values larger than 2.
Furthermore, based on the sum of the DFS of all observation types
assimilated in TOPAZ, we can estimate the relative impact of the $j^{th}$ type
of observations (RDFS$_j$):
$$\text{RDFS}_j = \frac{\overline{\text{DFS}}_j}{\sum_{l=1}^{O}\overline{\text{DFS}}_l}\times 100\%, \qquad (9)$$
where $O$ is total number of observation types. Figure 12 shows the
relative contribution of each observational data set in March. As
expected, the assimilation of ice concentration dominates the total DFS,
while the impacts of SST and SLA are limited to the region that are not
ice covered. Profiles of ocean temperature and salinity near the North
Pole in the Arctic are collected by the Ice-Tethered Profiler Program
(Krishfield et al., 2008; Toole et al., 2011). They have a very large impact
but they are very sparse. In March the SMOS-Ice data has a significant
impact (> 20 % of the total DFS) in the Northern Barents Sea, the
Western Kara Sea, Baffin Bay, the Greenland Sea and in Hudson Bay. In
November, the relative contribution is still significant in the Barents Sea,
the Kara Seas and in the Greenland Sea, but it is also significant in the
Beaufort Sea and in the Canadian Archipelago.

# 6.    Summary and Discussion

The thickness observations of thin sea ice in the Arctic can be derived from SMOS brightness temperature at 1.4 GHz (Tian-Kunze, et al., 2014; Kaleschke et al., 2016). This data set is available in near real time since 2010 at daily frequency. The study in this paper investigates the impact of assimilating this data set within the TOPAZ system, which is the Arctic component of the Copernicus Marine Services. It is shown that for thin ice (less than 0.4 m), the TOPAZ reanalysis and the SMOS-Ice have comparable distributions (though TOPAZ slightly overestimates the thin ice thickness from January to April) and that conditions are favorable for assimilating this data set.

We investigate the impact of assimilating SMOS-Ice (thinner than 0.4 m) in TOPAZ that already assimilates ice concentration, ice drift, SST, SLA and temperature and salinity profiles. The comparison is carried out for two periods: February-March and October-November of 2014. The study shows that the assimilation of SMOS-Ice data reduces the thickness RMSD of thin sea ice in March and in November by about 11% and 22% respectively, mainly caused by the reduction of the bias (too thick sea ice that seems larger in 2014 than in previous years). There are also some small improvements for SIC. The RMSDs for SST and SLA remain unchanged but are not degraded.

When compared to independent observations of SIT (IMB buoys) and sea ice draft (BGEP moorings) it is found that assimilation of SMOS-Ice yields improvements near the ice edge next to where SMOS-Ice has been assimilated but does not lead to improvements nor degradations in the rest of the Arctic.

 In this study, the DFS is used to evaluate the relative contributions of assimilated observations to the reduction of error in the TOPAZ system. The SMOS-Ice data have a smaller impact than ice concentration, but it has a significant contribution (defined as larger than 20 % of the total impact from all observations) in some areas; namely in the Greenland Sea, the Kara Sea, the Barents Sea, the Baffin Bay and the Hudson Bay

in March and in the Greenland Sea, the Kara Sea, the Barents Sea, the Beaufort Sea and the Canadian archipelago in November.

These studies follow from the first attempt of assimilation of SMOS-Ice with the LSEIK in a regional MITgcm configuration (Yang et al. 2014). Compared to this study, it is found that assimilation of SMOS-Ice has a more moderate impact. This may be related to the fact that TOPAZ uses a more complete observation network and that the assimilation has been spun up over a longer period of time (from 1989). We also find that assimilation of SMOS-Ice is comparatively larger in October-November than in February-March the time period when Yang et al. (2014) tested assimilation of SMOS-Ice. We also verified that assimilation of SMOS-Ice does not degrade ocean variables (SST and SLA), which could happen with a strongly coupled data assimilation scheme. Finally, we quantified the relative influence of SMOS-Ice for constraining the mode of variability in TOPAZ compared to a standard observation network.

To conclude, our study suggests that SMOS-Ice can be assimilated without degradation of other skills in our operational forecasting system. The benefits are generally small but can be significant for some regions near the ice edge. However, further work needs to be done to better understand the uncertainty of the assimilated SIT from the SMOS-Ice. Recently, Yang et al. (2016) tested the sensitivity of assimilating the SMOS-Ice data with the LSEIK during the winter of 2011-2012, and found that perturbations of the atmospheric forcing is important for improving the performance of assimilation, in agreement with Lisæter et al. (2007).

In the future, we may use the "saturation ratio" that is defined by the relationship of the variable L-band penetration depth and the maximal retrieval thickness as a function of temperature and salinity with which we can better identify the valid observations of sea ice thickness from SMOS. In addition, the satellite CryoSat-2 provides freeboard height data in thick ice that can complement the observations from SMOS (Kaleschke et al., 2010). The new sea ice thicknesses derived from a combination of SMOS and CryoSat-2 will be soon available (Kaleschke et al., 2015). Incidentally, the U.S Navy Arctic Cap Nowcast/Forecast System (ACNFS) is currently

testing the assimilation of a combined sea ice thickness product (personal communication from David Hebert) where the sea ice thickness is blended from SMOS-Ice and CryoSat-2 based on each satellite retrieval error.

## **Acknowledgment**

The authors are grateful to two anonymous reviewers and Jennifer Hutchings for their insightful comments that were helpful in improving the paper. Thanks to Dr. Y. Wang for useful discussions. We thank to the US National Snow and Ice Data Center (NSIDC) for providing the IceBridge data. This study was supported by ESA contracts 4000101476/10/NL/CT and 4000112022/14/I-AM and CPU time from the Norwegian Supercomputing Project (NOTUR II grant number nn2993k).

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

| Type | Spacing | Resolution | Provider |
|------|---------|------------|----------|
| SLA | Track | - | CLS |
| SST | Gridded | 5 km | OSTIA from UK Met Office |
| In-situ T | Point | - | Ifremer + other |
| In-situ S | Point | - | Ifremer + other |
| SIC | Gridded | 10 km | OSISAF |
| Ice drift | Gridded | 62.5 km | OSISAF |

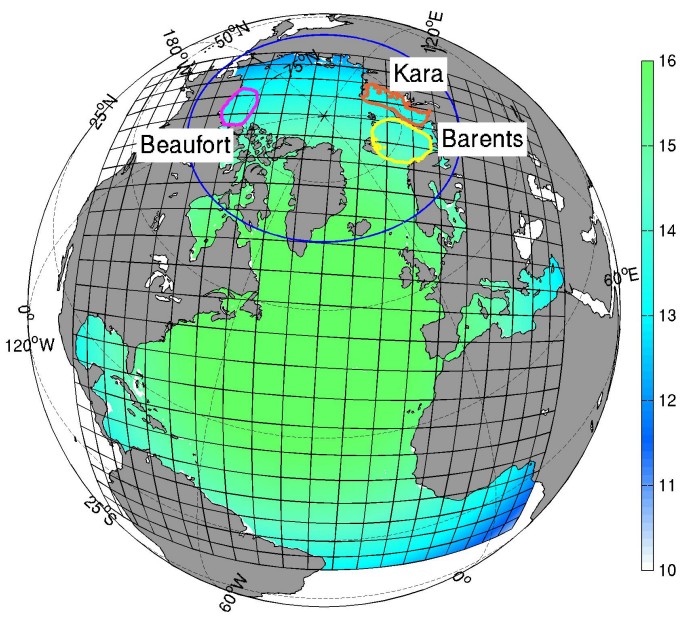

**Fig**. 1 TOPAZ model domain and horizontal grid resolution (km) with color shading. The blue line delimits the Arctic region (north of 63°N) and other color lines delimit the three marginal seas discussed in this study.

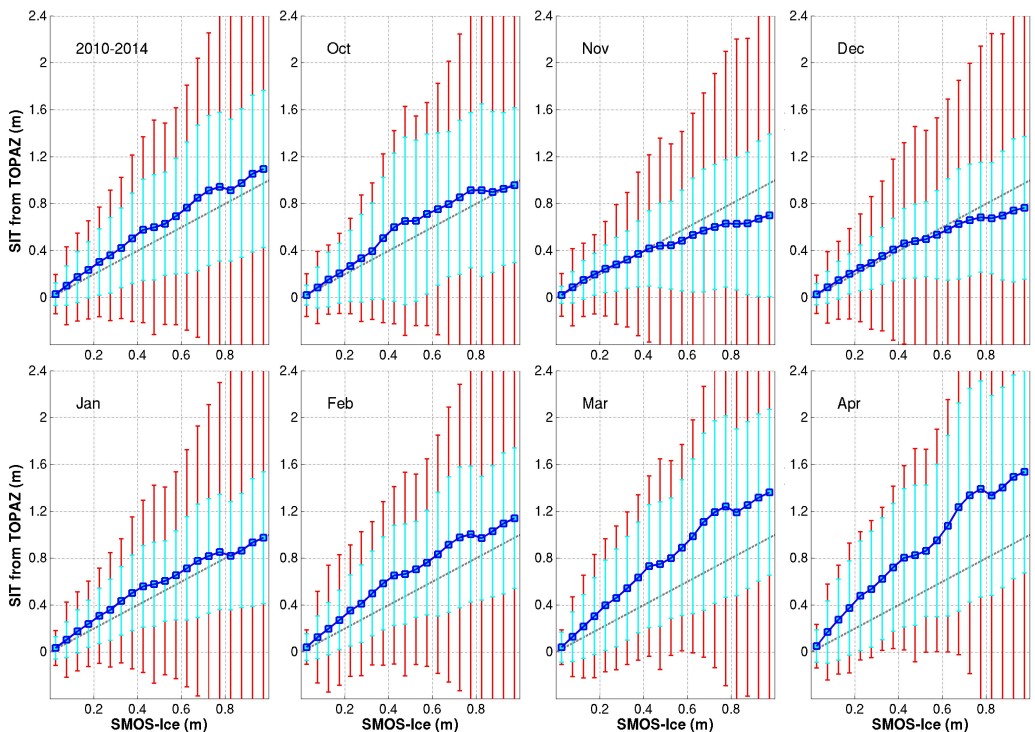

**Fig**. 2 Conditional expectations of TOPAZ versus SMOS-Ice (with bin of 5 cm) for each month calculated over the period 2010-2014. The cyan error-bars correspond to the RMSD against observations within each bin. The red error-bars correspond to the averaged standard deviations of observation error. The gray dashed line denotes the line y=x.

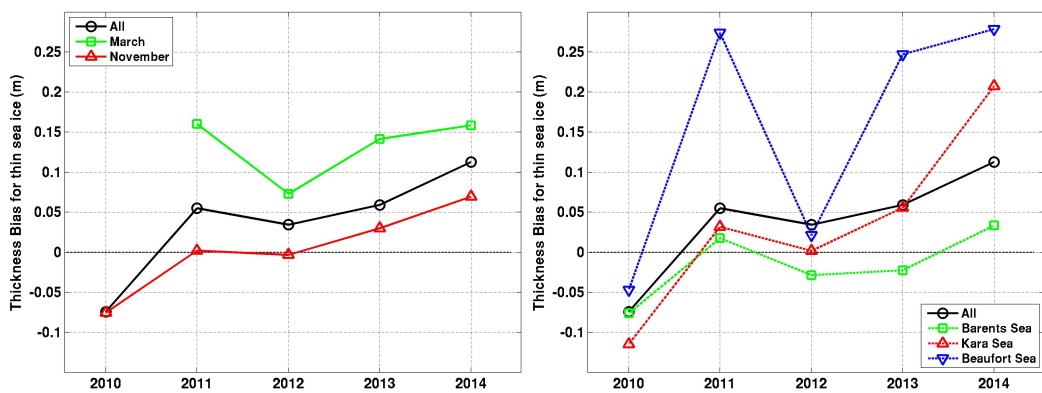

**Fig**. 3 Yearly thickness biases of thin sea ice from TOPAZ compared to SMOS-Ice observations (Eq. 4). The black line represents the yearly mean bias. **Left**: the green (red) line represents the mean bias for March (November) months. **Right**: the colored lines represent the biases in the Barents Sea, the Kara Sea, and the Beaufort Sea.

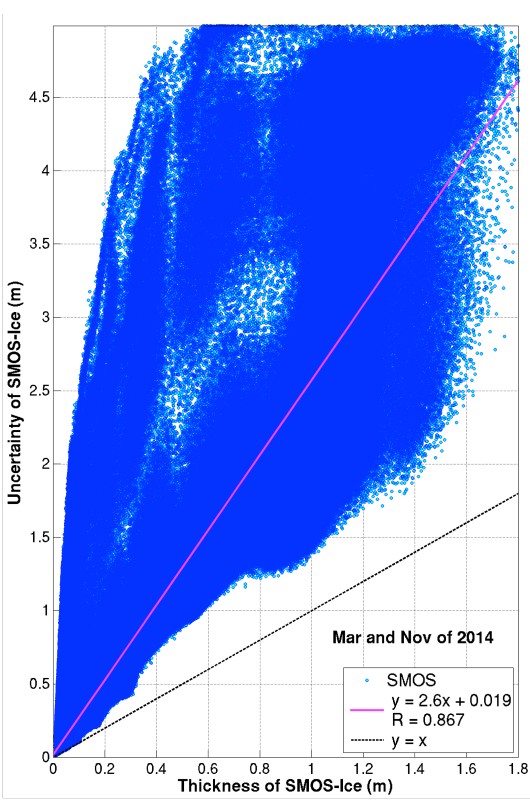

**Fig. 4** Scatter plot of the uncertainty of the observation as function of the observed thickness from SMOS in March and November of 2014.

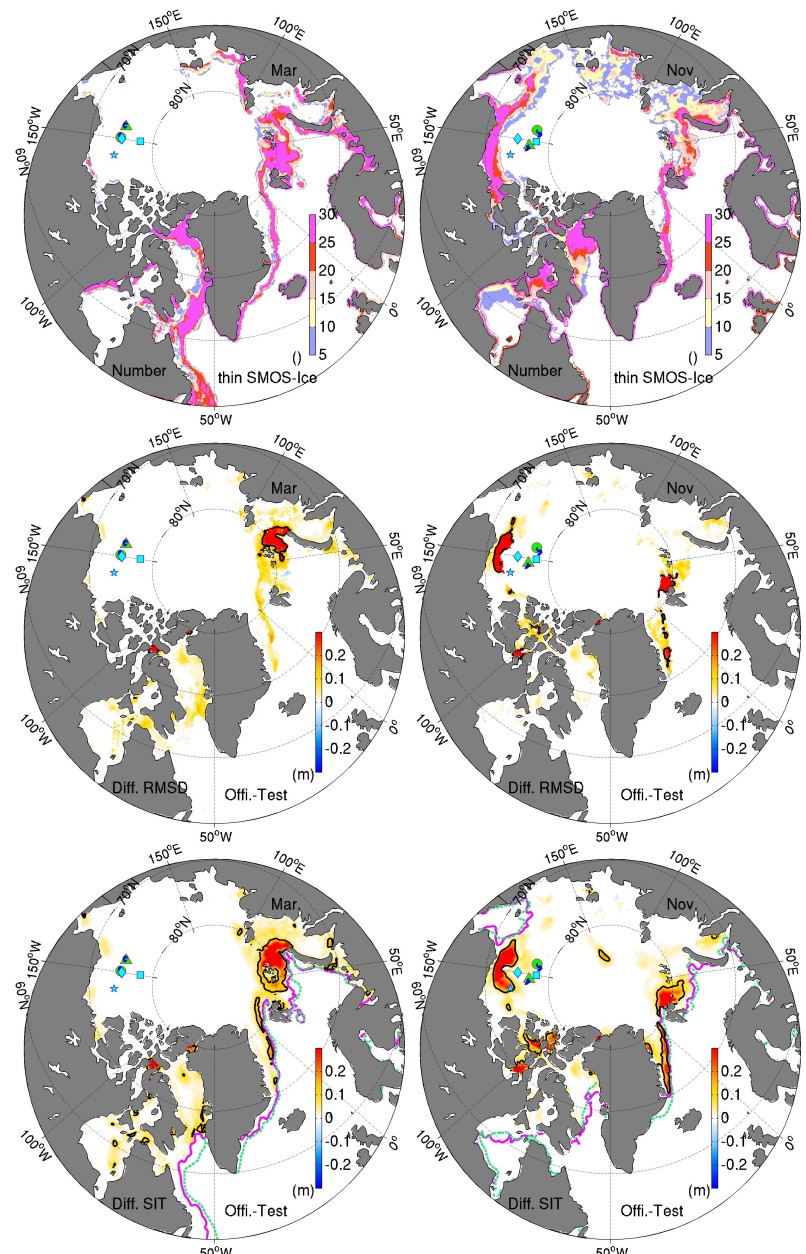

**Fig**. 5 **Top Row**: Number of the valid SMOS-Ice data in March (left) and in November (right) of 2014. The trajectories of the buoys *2013F* and *2013G* (*2013F* and *2014F*) from IMB are the blue lines in March (November). Their first positions are marked by circle and triangle respectively. In March (November), the mooring locations from BGEP - *2013a*, *2013b*, and *2013d* (*2014a*, *2014b*, and *2014d*) - are marked by diamond, square and pentagram respectively. **Middle Row**: Difference of RMSDs for the thin SIT between Official Run and Test Run. The black line denotes the 0.2 m isoline. **Bottom Row:** Difference of SIT between Official Run and Test Run. The black line denotes the 0.2 m isoline, the green (magenta) line is the 15% concentration isoline from OSISAF (Official Run).

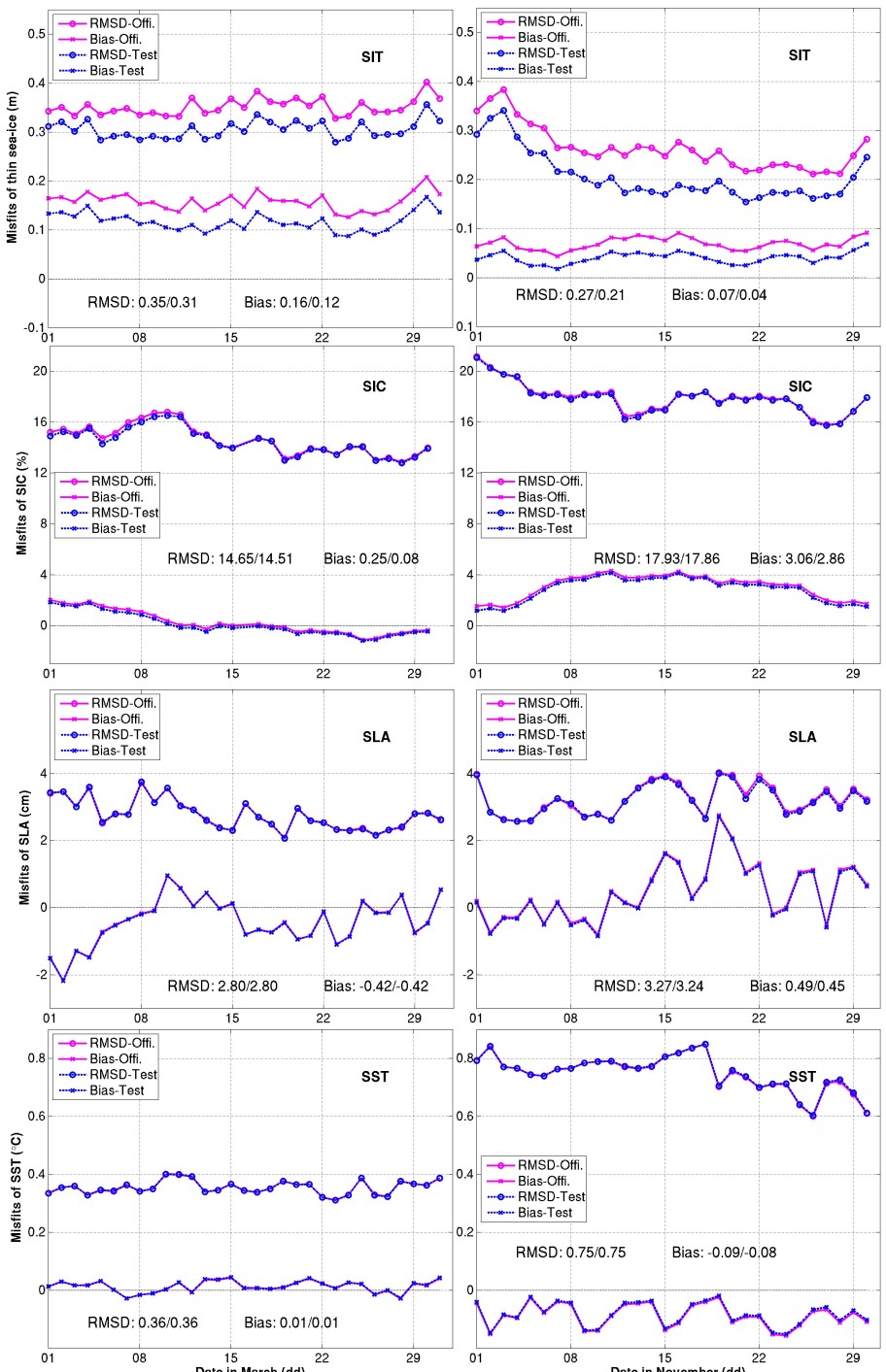

**Fig**. 6 Daily time series of the bias (marked with crosses) and the RMSD (marked with circles) calculated for the Arctic region in the Official Run (magenta) and the Test Run (blue) for different variables in March (Left) and November (Right).

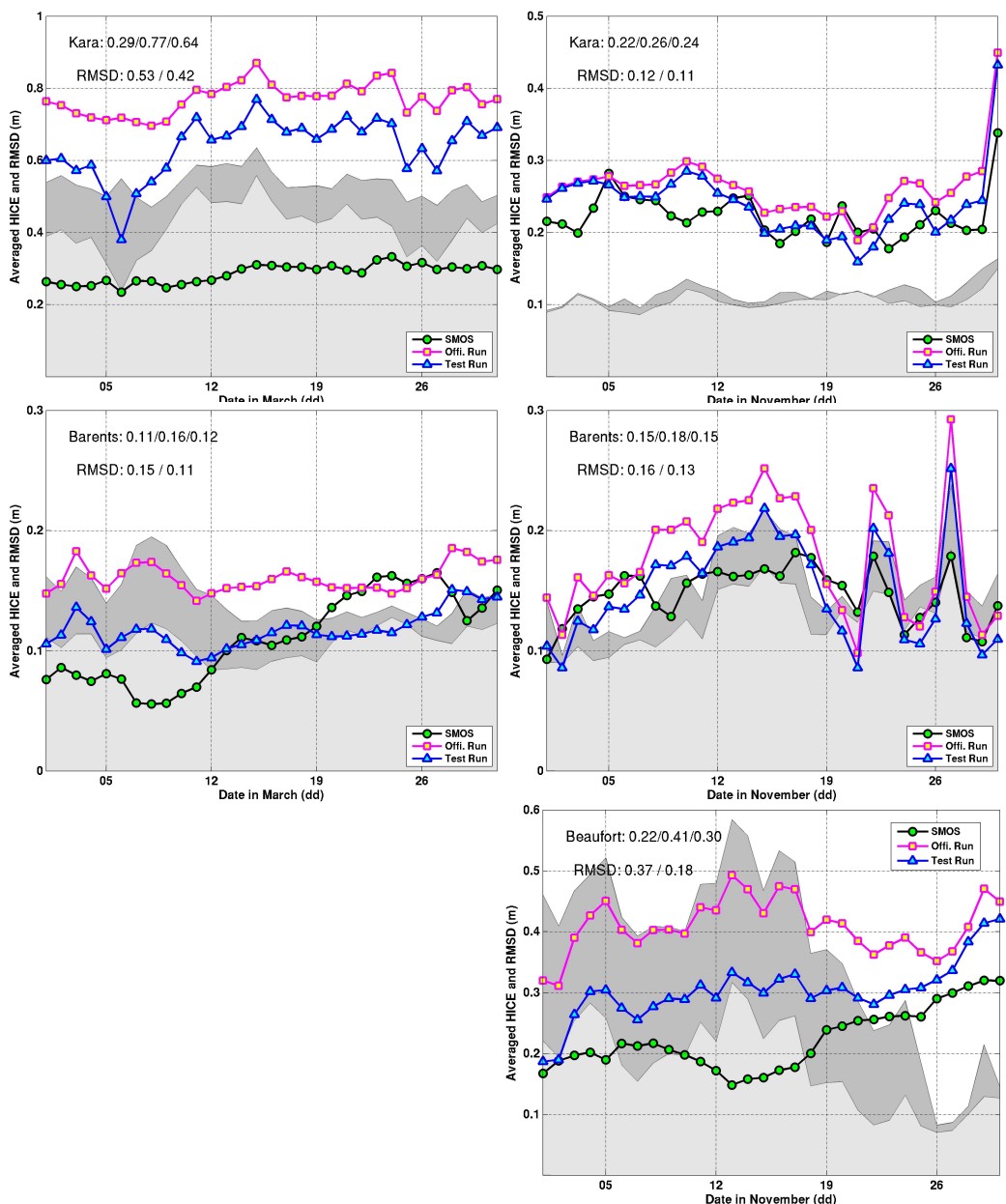

**Fig**. 7 Daily time series of the mean SIT for thin sea ice in the Kara Sea (top row), the Barents Sea (middle row) and Beaufort Sea (bottom row) in March (*left*) and November (*right*). The light (dark) gray shading is the daily spatial RMSD of thin sea ice in the Test Run (Official Run).

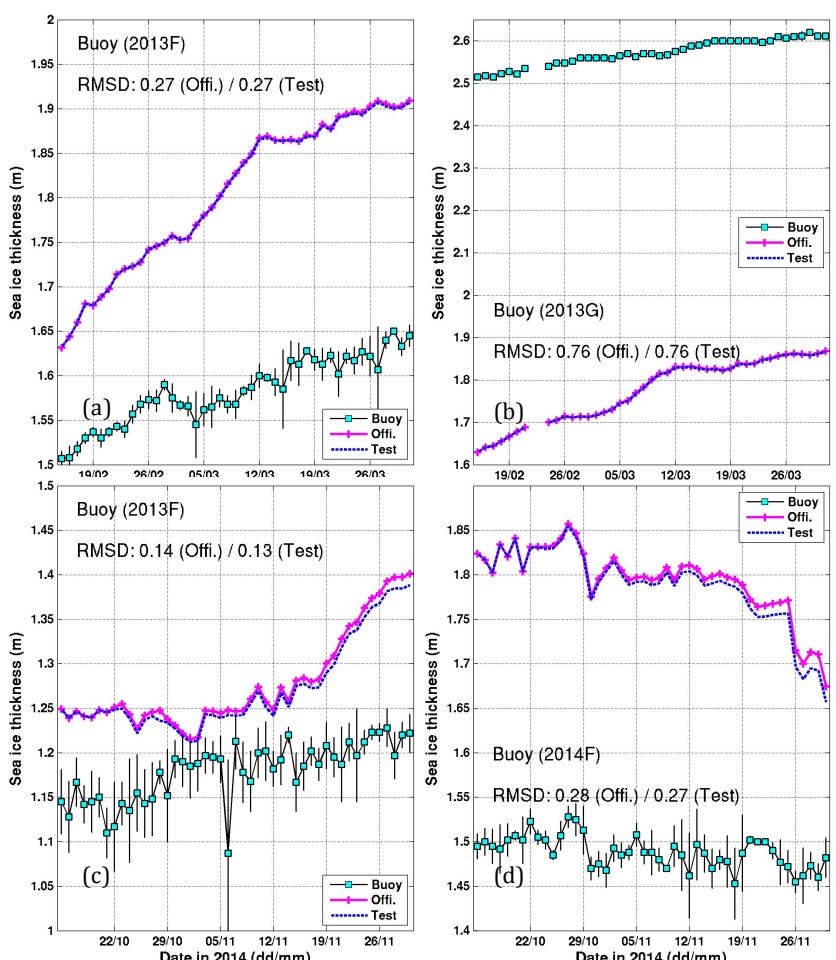

**Fig 8.** Daily time series of SITs from Official Run (crossed magenta line) and Test Run (dashed blue line) compared to the buoy measurements from IMB (squared black line). The daily standard deviations of the observations are shown with error bars. The buoy locations and their drift trajectories in the month are shown in **Fig**. 5. **Upper row** covers the period 15[th] Feb to 30[th] Mar 2014 by (a) *Buoy 2013F* and (b) Buoy *2013G*. **Bottom row** covers period 15[th] October to 30[th] Nov 2014 by (c) Buoy *2013F* and (d) Buoy *2014F*.

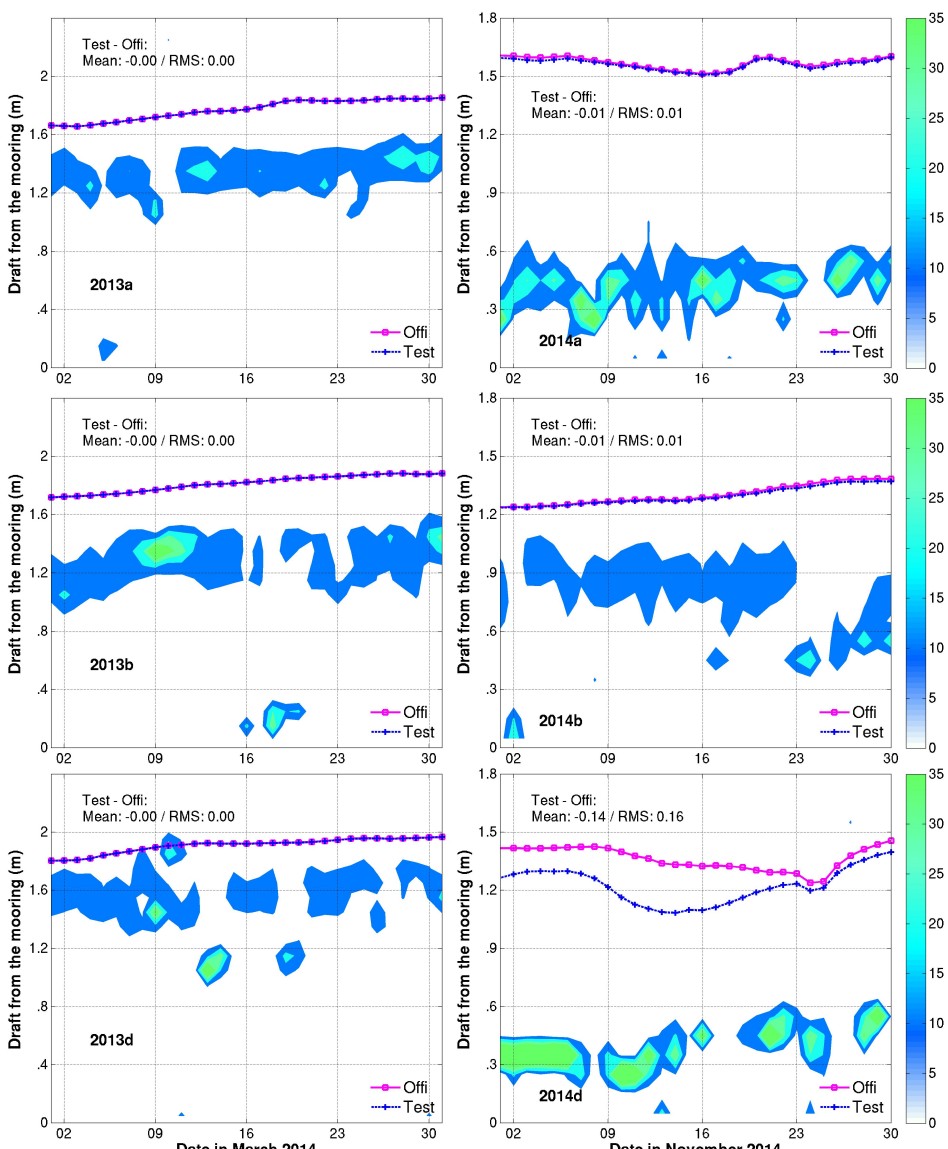

**Fig**. 9 Comparison of sea ice drafts from the Official Run (squared-magenta line), the Test Run (dashed-blue line) and from the bottom-tethered moorings of BGEP. The left (right) panels are for March (November) 2014. The daily histograms of sea ice draft (frequency percents for 0.1 m bins) are shown with shading colors. The positions of the moorings are marked in Fig. 5.

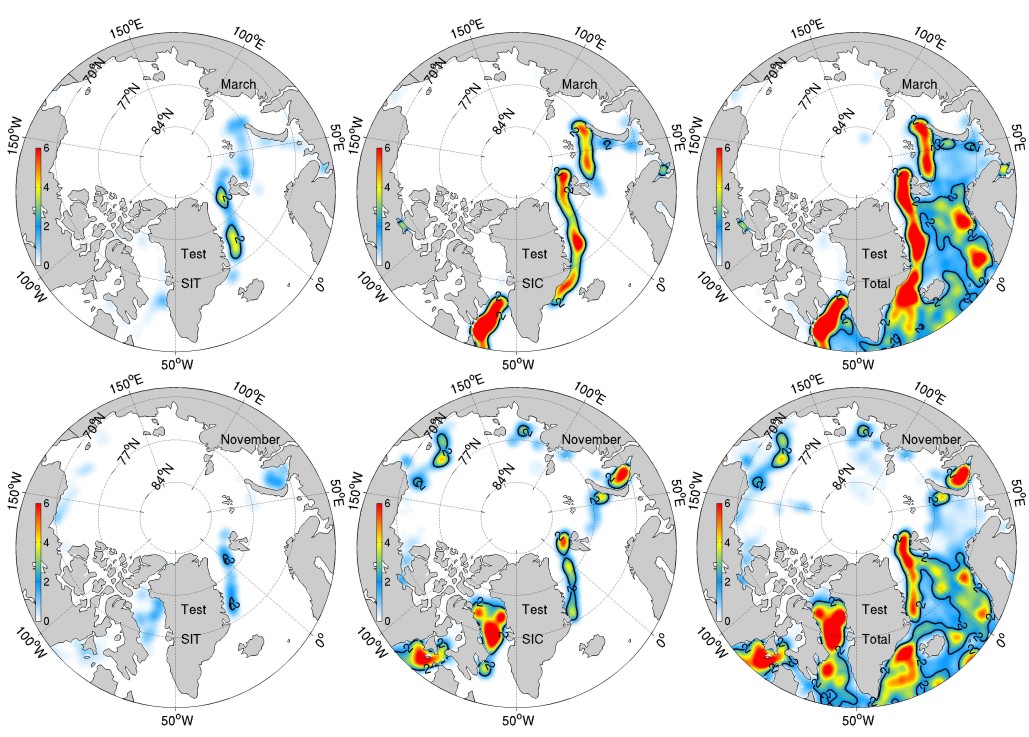

**Fig**. 10 Monthly averaged DFS from the Test Run in March (*upper*) and in November (*lower*) for sea ice thickness from SMOS-Ice (left column), sea ice concentration from OSISAF (middle column), and the total DFS of all assimilated observations (right column). The black line denotes the isoline of DFS equal to 2.

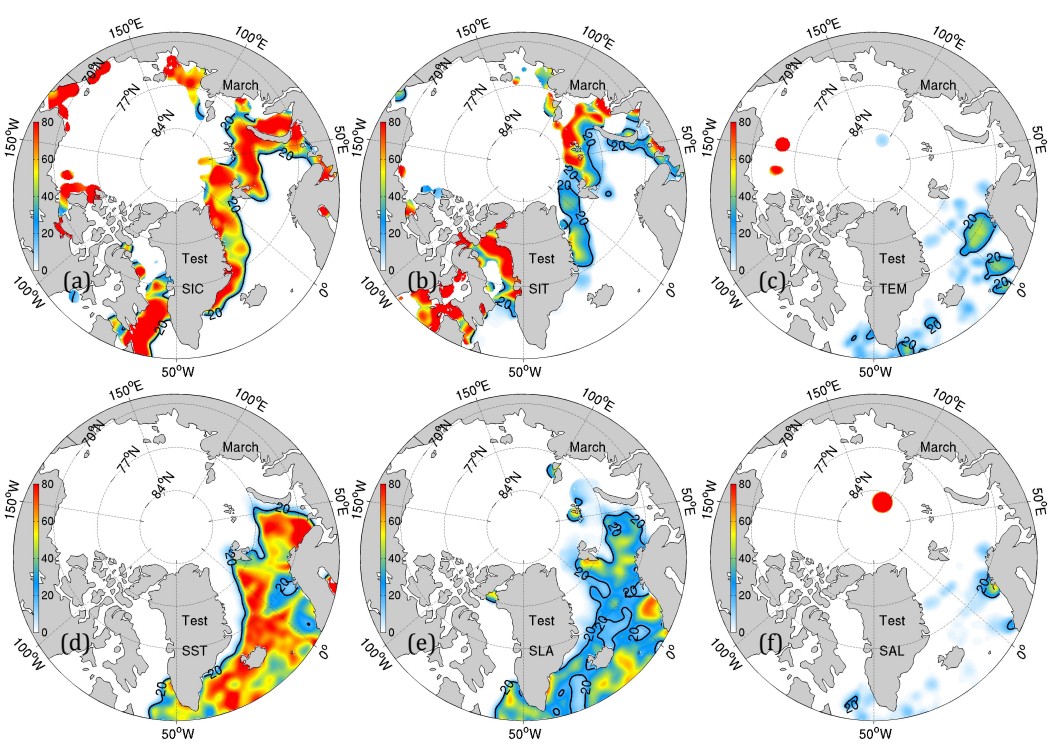

**Fig**. 11 Relative contributions of each observational data set in the total DFS during March 2014. Panel (a) is for sea ice concentration from OSISAF; (b) sea ice thickness from SMOS-Ice; (c) temperature profiles; (d) SST; (e) along-track Sea Level Anomaly; (f) salinity profiles. The black line is the 20% isoline.

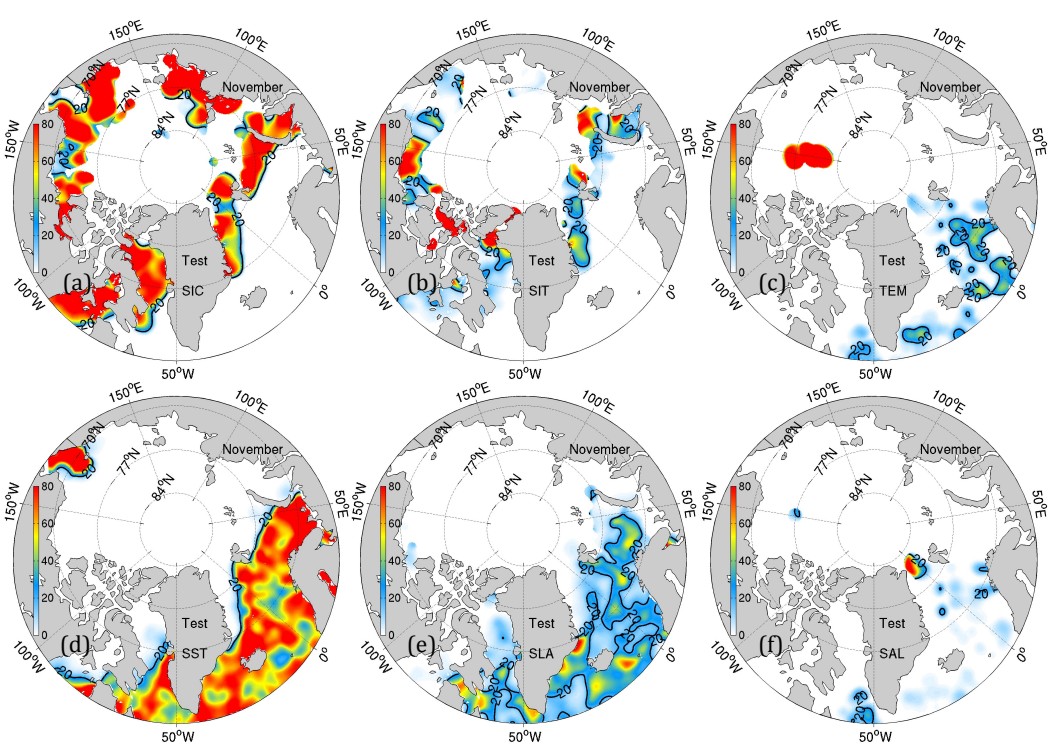

**Fig**. 12 Same as Figure 11 for November 2014