# Peer review of "Benefits of assimilating thin sea ice thickness from SMOS into the TOPAZ system"

_The Cryosphere, 2016_

## Referee Comment (RC1) · Anonymous Referee #1 · 7 Jul 2016

Review of "Benefits of Assimilating thin sea-ice thickness from SMOS-Ice into the TOPAZ System", J. Xie, F. Counillon, L. Bertino, X. Tian-Kunze, and L. Kakeschke.

General Comments

This paper examines the assimilation of SMOS derived ice thickness into the TOPAZ modeling system. Assimilating ice thickness is a timely subject, and very much needed to help constrain model runs, especially short term forecast systems. Two modeling experiments were performed spanning Feb-Mar and Oct-Nov 2014, with and without SMOS thickness. Comparisons to daily SMOS thickness values are made. Also, a degree of freedom test is performed to investigate the impact SMOS ice thickness has on the model data.

Overall, I found this paper somewhat lacking in quality. There are several instances

where terms are not defined and instances where the same mathematical symbol is used for different terms, making some of the equations confusing to follow. There are also many places where key details are omitted. Therefore, I recommend the paper be returned and re-submitted after major revisions.

Specific Comments

Page 3, line 22: "Measurements of thick sea ice draft. . ." should read "Measurements of thick sea ice freeboard. . ." as altimeters measure freeboard, not draft.

Page 5, line 2: the authors state the thickness of TOPAZ was validated over the period 1991-2013 using ICESat and IceBridge. ICESat was from 2003-2009, and IceBridge started after that. What data was used starting in 1991 to validate the model? They refer to an unpublished manuscript submitted by Xie (2016) several times. I find this troubling, as it is not peer-reviewed and cannot be referred to. It also does not say where is was submitted. I suggest a different reference be used throughout the manuscript.

Page 5, line 4, the authors state "While the spatial pattern and regression compare reasonably well, large biases exist" What regression and spatial pattern are they talking about? Are the biases positive or negative? After talking about TOPAZ validation, they state inaccuracy in the ice thickness is a drawback. . .. More detail needs to be added to this section.

Page 5, line 31: Is the ice model a multi-category model or one layer? This is important because it will come into play when assimilating ice thickness.

Page 6, equation 3 and lines thereafter: The $P^a$ covariance is described, but not used. I'm not sure what $P^a$ is supposed to be. Also, on line 24 the authors state "the extra term is quadratic and positive." What term is extra, the second term?

Page 7, line 8: Another reference to unpublished/unaccepted manuscript for validation. This publication needs to be accepted first, or include the details from that validation in this paper.

Page 7, line 12: where are the SMOS-Ice products available? A reference or website should be included. Is it available in near real time for operational centers?

Page 7, line 15: What does "TOPAZ equivalent" mean? Is there some spatial averaging or processing to match the observation location for comparison?

Page 7, line 17: The term 'RMSD' is used, but not defined until Eq. (6) on page 9. RMSD should be defined here.

Page 8, line 16: This is the first time the term "innovations" is used. I presume this is referring to the 2nd term on the RHS of equation (1). This should be spelled out.

Page 8, equation (4): y_smos is not defined. I guess this is the SMOS observation thickness. Also, is sea ice volume assimilated into the sea ice model? Is there only one ice category in their model to assimilate volume? If more than one category, how do the authors decide what category to assimilate the thickness?

Page 8, line 31. The authors state they implement an upper limit on observation standard deviation of 5 meters. This seem like a large standard deviation value (12.5 times the max observation value of 0.4 m) given they are only assimilating SMOS observations up to 0.4 meters. Why was 5 meters chosen for the standard deviation limit? Are there SMOS observations with 0.5 meters with a 5 meter standard deviation? Figure 2 suggests standard deviations less than 2 for SMOS < 0.4 meters.

Page 8, line 25: why use the symbol TSLA? SLA is used for along-track sea level anomaly on page 4 and 6.

Page 9, line 10: Sea ice thickness SMOS-Ice is stated to be in Table 1, but it is not listed there.

Page 9, equations 5 and 6: What is 'H'? It is used as Bilinear operator and Obs error in previous equations.

Page 9, line 20: I find it interesting that there are minimal observations in the Beaufort

during March. Can the authors expand on why this is? Is the ice too thick for SMOS at this time? Or not there?

Page 10: line 15: The terminology "highlighted with marked lines" is confusing to me. I think it would be clearer if the authors state something like "the averaged thickness of thin sea-ice . . . are shown with marked lines in the panels of Figure 6".

Page 11, equation (7): does 'tr' mean trace? This, as well as all terms, should be defined to ensure clarity.

Page 11, line 25: On this line I think the authors are referring to equation (7).

Page 11, equation (8): This looks more like a root mean square (RMS) than a mean. Is there a reason why the authors decided to use a RMS here? Also, what are the subscripts i and j?

Page 12: equation (9), what is subscript j? j'th observation set?

Page 12: line 6: Is there a reference for the ice-tethered profile data? Is this ice or ocean profile data?

Page 13: line 30: I would say the blended sea ice thickness has been "tested with" the U.S. Navy Arctic Cap Nowcast/Forecast system. The term "implemented" implies the blending is currently being used operationally, which is not the case.

Figure 6: If I understand correctly, the blue line with triangles is the test run where ice thickness is assimilated once a week. There does not seem to be any evidence of the assimilation. I would expect the blue triangle line to get closer to the SMOS line at the assimilation interval. Why does this not occur?

Technical Corrections

In many places throughout the document (e.g., page 2 lines 13 and 18, page 3 line 16, page 4 line 19, ) words are run together without proper spacing. I don't know if this is an artifact of the submission formatting or something else, but the word and sentence

spacing needs to be verified.

Page 3, line 29: "Cryostat-2" should be "CryoSat2"

Page 4, line 13: Yang reference is (2015), but in bibliography it is 2014. Please check on the year of this publication.

Page 4, line 14: Define LSEIK.

Page 4, line 16: should read "This study is a follow up and assesses. . ."

Page 4, line 16: What is this a follow-up of? Yang (2015)?

Page 4, line 24: should read ". . . and does not apply post processing. . ."

Page 8 and after: no need to bold "official run" and "test run". I find this distracting in the rest of the document.

Page 12, line 25: need units after 0.4 (should read thinner than 0.4 m)

Page 12, line 32: Yang reference is (2015), but in bibliography it is 2014. Please check on the year of this publication.

Figure 1: the words of the regions are hard to see (especially Kara and Beaufort). I suggest putting a white background for these words.

Figure 3: Don't need the word "resp" when doing comparisons. Usually when comparisons are done in this manner you just state the contrasting item. The green (red) line represents the mean bias for March (November) of each year.

Figure 4: Bottom row. It is hard to see the orange line. I suggest choosing a separate color not in the colorbar. Same comment about "resp" as above. Say Top Row, Middle Row, Bottom Row.

Figure 5: The vertical axis changes on each plot. Please plot each item with the same vertical axis.
Figures 6, 7, 8: same comment about "resp" and bold test run, official run.

[Figure]

---

## Referee Comment (RC2) · Anonymous Referee #2 · 13 Jul 2016

General Comments

The paper investigates the impacts of assimilating SMOS sea ice thickness into the TOPAZ system. The DA experiment of assimilating SMOS sea ice thickness is compared with the experiment without SMOS thickness. It shows that assimilating SMOS thickness can reduce the thin sea ice thickness errors (as expected), slightly improve the sea ice concentration, but does not degrade other ocean variables. Further, the DFS method is used to quantify the impact of SMOS ice thickness on the model data. Overall, the result is encouraging, and would be helpful for future sea ice prediction and reanalysis efforts. However, some issues need to be further addressed. I recommend that this manuscript be accepted after a major revision. The following comments/suggestions are provided for the authors to consider.

Specific Comments/suggestions:

(1) Page 2, line 6: "winter season" should be "cold season".

(2) Page 2, line 13, the full name of TOPAZ should be given in the Abstract.

(3) Page 2, line 20, should "contents" be corrected to "contains" ?

(4) Page 2, line 28, the "Keywords" should be revised, e.g., a lot of readers do not know "OSE" and "DFS".

(5) Page 3, line 22, "draft" should be "freeboard".

(6) Page 4, line 13: Yang et al. (2015) should be (2014).

(7) Page 4, line 14: "LSEIK" should be defined.

(8) Page 4, line 26, "Xie et al., 2016" is frequently referred in this MS, this should be corrected, as it has not been accepted, the authors even have not tell us the journal they submitted to.

(9) Page 7, line 15, is "TOPAZ equivalent ice thickness" "TOPAZ model mean ice thickness"?

(10) Page 7, line 17, "RMSD" is not defined here.

(11) Page 7, line 29, you only assimilate the SMOS data less than 0.40 m, why not 0.50 m? As you referred, "the penetration depth into sea ice is about 0.5 m". Although you mentioned that "the effect of ice melting may lead to a saturation thickness of less than 0.4 m", but for this paper, you run the experiments in the cold season, basically there is no melting in the sea ice surface. If you increase the upper limit, more SMOS observation data is available, thus stronger influence/correction to the TOPAZ system is expected. In Yang et al. (2014), they use an upper limit of 1.0 m.

(12) Page 8, line 3, 4, 7: "thick" should be "thickness"?

(13) Page 8, line 19, ysmos is not defined.
(14) Page 9, line 10: "SMOS-Ice" is forgotten in Table 1.

(15) Page 11, line 1: In the Beaufort Sea, there are some sea ice draft measurements from Beaufort Gyre Exploration Project (BGEP) by upward-looking sonar (ULS) moorings located in the Beaufort Sea (http://www.whoi.edu/beaufortgyre). Also, there are some sea ice thickness data obtained from autonomous ice mass balance (IMB; http://imb.erdc.dren.mil ). I would suggest the authors to use these data as the independent ice thickness observations in the evaluation of their model results.

(16) Page 12, line 32, an "a" is missing before "slight".

(17) Page 13, line 24, should be "In addition".

---

## Author Response (AR1)

**Responses to reviewers' comments**

We would like to thank the reviewers for their careful readings of our manuscript and the detailed comments. The major changes in the revision are the following:

- Added comparison with the independent sea ice thickness from buoy observations from autonomous ice mass balance (IMB; http://imb.erdc.dren.mil) shown in Fig. 7 of the revised manuscript.
- Equations from (1) to (5) are modified or changed their orders in the text.
- Modifications of figures of 1, 4, and 5 as the suggestions.

The detailed responses are listed one by one as following:

**Reviewer #1:**

*Page 3, line 22: "Measurements of thick sea ice draft. . ." should read "Measurements of thick sea ice freeboard. . ." as altimeters measure freeboard, not draft.*

**Reply**: Thank you, this is now corrected.

*Page 5, line 2: the authors state the thickness of TOPAZ was validated over the period 1991-2013 using ICESat and IceBridge. ICESat was from 2003-2009, and IceBridge started after that. What data was used starting in 1991 to validate the model? They re- fer to an unpublished manuscript submitted by Xie (2016) several times. I find this trou- bling, as it is not peer-reviewed and cannot be referred to. It also does not say where is submitted. I suggest a different reference be used throughout the manuscript.*

**Reply**: We have clarifed that. Xie et al. (2016) was submitted to *Ocean Science*, and is available online as doi:10.5194/os-2016-38. In this paper, the reanalysis for the period 1991-2013 is validated with in situ data and satellite data; namely ICESat (2003-2008), IceBridge (2009-2011), and other in situ data for the period 1993-2005 (Lindsay, 2013).

*Page 5, line 4, the authors state "While the spatial pattern and regression compare reasonably well, large biases exist" What regression and spatial pattern are they talking about? Are the biases positive or negative? After talking about TOPAZ validation, they state inaccuracy in the ice thickness is a drawback. . .. More detail needs to be added to this section.*

**Reply**: Thanks. This is corrected in the revision.

"In the Arctic reanalysis, the daily sea ice thickness of TOPAZ has been validated for the period 1991-2013 compared to different types of available observations (Xie et al., 2016). TOPAZ shows good agreement with the spatial distribution of ice thickness in ICESat data (available between 2003 and 2008) with a spatial correlation 0.74 in spring and 0.84 in autumn. However, TOPAZ shows a clear overestimation of ice thickness in the Beaufort Sea and an underestimation in the other areas of the Arctic."

*Page 5, line 31: Is the ice model a multi-category model or one layer? This is important because it will come into play when assimilating ice thickness.*

**Reply**: The sea ice model has only one layer (two category, ice or no ice). This is clarified in the model description.

"The NERSC-HYCOM model is coupled to a one thickness category sea-ice model for which the ice thermodynamics are described in Drange and Simonsen (1996) and the ice dynamics are based on the elastic-viscous-plastic rheology described in Hunke and Dukowicz (1997) and with a modification from Bouillon et al. (2013)."

*Page 6, equation 3 and lines thereafter: The $\hat{P}^a$ covariance is described, but not used. I'm not sure what $\hat{P}^a$ is supposed to be. Also, on line 24 the authors state "the extra term is quadratic and positive." What term is extra, the second term?*

**Reply**: $P^a$ is the residual error covariance, posterior to the assimilation of data. It is indeed not explicitly required in the paper. We have revised this part of the manuscript to clarify the particularities of the DEnKF.

*Page 7, line 8: Another reference to unpublished/unaccepted manuscript for validation. This publication needs to be accepted first, or include the details from that validation in this paper.*

**Reply**: The paper by Xie et al. (2016) is currently under review for ocean science but available in ocean science discussion.

Xie, J., Bertino, L., Counillon, F., Lisæter, K. A., and Sakov, P.: Quality assessment of the TOPAZ4 reanalysis in the Arctic over the period 1991–2013, Ocean Sci. Discuss., doi:10.5194/os-2016-38, in review, 2016.

*Page 7, line 12: where are the SMOS-Ice products available? A reference or website should be included. Is it available in near real time for operational centers?*

**Reply**: The information is now added.

Page 7 line 9-13: "They are provided by Hamburg University at the website of https://icdc.zmaw.de/1/daten/cryosphere/l3c-smos-sit.html (Kaleschke et al., 2012; Tian-Kunze et al., 2014). SMOS sea ice thickness maps are provided at daily frequency from October 2010 and are available in near-real time during the cold season."

*Page 7, line 15: What does "TOPAZ equivalent" mean? Is there some spatial averaging or processing to match the observation location for comparison?*

**Reply**: Sorry this was unclear. The sentence is replaced by "The TOPAZ ice thicknesses shown in Fig.2 are at the same locations and times as the observations."

*Page 7, line 17: The term 'RMSD' is used, but not defined until Eq. (6) on page 9. RMSD should be defined here.*

**Reply**: Thanks. The definition of the bias and the RMSD are moved up.

*Page 8, line 16: This is the first time the term "innovations" is used. I presume this is referring to the 2nd term on the RHS of equation (1). This should be spelled out.*

**Reply**: Thanks. The definition of "innovations" is added in Section 2.2.

*Page 8, equation (4): y_smos is not defined. I guess this is the SMOS observation thickness. Also, is sea ice volume assimilated into the sea ice model? Is there only one ice category in their model to assimilate volume? If more than one category, how do the authors decide what category to assimilate the thickness?*

**Reply**: The definition of $y_{smos}$ is added. Our sea ice model only has one category of ice thickness.

*Page 8, line 31. The authors state they implement an upper limit on observation standard deviation of 5 meters. This seem like a large standard deviation value (12.5*

*times the max observation value of 0.4 m) given they are only assimilating SMOS observa- tions up to 0.4 meters. Why was 5 meters chosen for the standard deviation limit? Are there SMOS observations with 0.5 meters with a 5 meter standard deviation? Figure 2 suggests standard deviations less than 2 for SMOS < 0.4 meters.*

**Reply**: A maximum observation error of 5 meters is set by default by the data provider for saturated values. These measurements are qualitative and cannot be assimilated without algorithmic developments. Other measurements may have uncertainties higher or lower than the observation values but there is nothing in the data assimilation framework that prevents from assimilating them. We show below the uncertainties of the observations as function of the observed thickness of SMOS-Ice in one month of March and November 2014 shown in the Fig. A. In March, the uncertainty is possibly close to 5 meter if the thickness thicker than 0.5 m. Meanwhile, for the thin sea ice (<0.4 m), the related uncertainty may be 10 times the observation value.

[Figure]

**Fig. A** The uncertainty of the observation as function of the observed thickness from the SMOS-Ice in March (left) and November (right) of 2014.

The text is modified as such p. 10, l. 9: "with an upper limit of 5 m beyond which the observations are assumed fully saturated".

*Page 8, line 25: why use the symbol TSLA? SLA is used for along-track sea level anomaly on page 4 and 6.*

**Reply**: Thanks. The inconsistencies are corrected, and we only use SLA in the paper.

*Page 9, line 10: Sea ice thickness SMOS-Ice is stated to be in Table 1, but it is not listed there.*

**Reply**: Thanks. The statement is changed now.

*Page 9, equations 5 and 6: What is 'H'? It is used as Bilinear operator and Obs error in previous equations.*

**Reply**: H is the observations operator, which computes the model equivalent of the observations, it is used for spatial interpolation with a bilinear operator as in equation (1), and the concerned statements are changed.

*Page 9, line 20: I find it interesting that there are minimal observations in the Beaufort during March. Can the authors expand on why this is? Is the ice too thick for SMOS at this time? Or not there?*

**Reply**: During March the observed sea-ice thicknesses are mostly thicker than 1 m in the Beaufort Sea, and the thin sea ice (< 0.4 m) appears around the Mackenzie estuary region, shown as the blue shading in the left of Fig. B. It implies the observed thicknesses have been rejected in the OSE runs. In addition, in the right panel of Fig. B the maximal uncertainties (about 5 m) occupy most of the Beaufort Sea, which may relate with the overestimation of the observation uncertainties in the version 2.1 of SMOS-Ice.

[Figure]

**Fig. B** Snapshot of Sea ice thickness (left) and its observation uncertainty (right) from the SMOS-Ice data in 15[th] March 2014. The dashed black line represents the domain of the Beaufort Sea.

*Page 10: line 15: The terminology "highlighted with marked lines" is confusing to me. I think it would be clearer if the authors state something like "the averaged thickness of thin sea-ice . . . are shown with marked lines in the panels of Figure 6".*

**Reply**: Thanks. It is replaced in the revision.

*Page 11, equation (7): does 'tr' mean trace? This, as well as all terms, should be defined to ensure clarity.*

**Reply**: Yes. The definition is added.

*Page 11, line 25: On this line I think the authors are referring to equation (7).*

**Reply**: This sentence was unclear and it has been revised now.

*Page 11, equation (8): This looks more like a root mean square (RMS) than a mean. Is there a reason why the authors decided to use a RMS here? Also, what are the subscripts i and j?*

**Reply**: Thanks for this comment. We use the mean DFS to replace the RMS of DFS in the revision, and also update the related figures and statements. In addition, $j$ represents the $j$'th type of the observations, $i$ accounts the times of data assimilation in the equation (8).

*Page 12: equation (9), what is subscript j? j'th observation set?*

**Reply**: Yes, $j$ represents the $j$'th type of the observation data-set assimilated in the TOPAZ system. The related statements in the revision are corrected.

*Page 12: line 6: Is there a reference for the ice-tethered profile data? Is this ice or ocean profile data?*

**Reply**: Yes, the related reference is added, and they are ocean profile data below the sea ice.

*Page 13: line 30: I would say the blended sea ice thickness has been "tested with" the U.S. Navy Arctic Cap Nowcast/Forecast system. The term "implemented" implies the blending is currently being used operationally, which is not the case.*

**Reply**: Thanks. The statement has been modified as recommended, and is changed as following:

"Incidentally, the U.S Navy Arctic Cap Nowcast/Forecast System (ACNFS) is currently testing the assimilation of a combined sea ice thickness product where the sea ice thickness is blended from SMOS-Ice and CryoSat2 based on each satellite retrieval error (personal communication from David Hebert)."

*Figure 6: If I understand correctly, the blue line with triangles is the test run where ice thickness is assimilated once a week. There does not seem to be any evidence of the assimilation. I would expect the blue triangle line to get closer to the SMOS line at the assimilation interval. Why does this not occur?*

**Reply**: The blue line is indeed from the test run. The reduction of error is not obvious unless one constructs a weekly cycle of the error but it is not necessarily visible if the error growth is slow. This is not only the case for SMOS but as well for other ocean and sea ice data types (see for example the skills of the operational runs on http://myocean.met.no/ARC-MFC/V2Validation/timeSeriesResults/index.html). The data assimilation updates are however visible as a little flashing if the daily outputs are animated, as can be done using the visualization service on http://marine.copernicus.eu. We are considering positively that the error time series does not bear the scars of assimilation steps as long as the error remains consistently below those of the official run. This means that data assimilation is keeping the test run constantly under control and that the forecast errors are unlikely to grow suddenly after assimilation.

*Technical Corrections*

*In many places throughout the document (e.g., page 2 lines 13 and 18, page 3 line 16, page 4 line 19,) words are run together without proper spacing. I don't know if this is an artifact of the submission formatting or something else, but the word and sentence spacing needs to be verified.*

**Reply**: Thanks. The related formatting is verified.

*Page 3, line 29: "Cryostat-2" should be "CryoSat2"*

**Reply**: Thanks. They are modified by CryoSat-2.

*Page 4, line 13: Yang reference is (2015), but in bibliography it is 2014. Please check on the year of this publication.*

**Reply**: Thanks. This mistake is corrected.

*Page 4, line 14: Define LSEIK.*

**Reply**: Yes, the definition is added.

*Page 4, line 16: should read "This study is a follow up and assesses. . ."*

**Reply**: Thanks. "The present study follows up the work from Yang et al. (2014) but uses a different model and assesses."

*Page 4, line 16: What is this a follow-up of? Yang (2015)?*

**Reply**: The rephrasing should have clarified the issue.

*Page 4, line 34: should read ". . . and does not apply post processing. . ."*

**Reply**: Thanks. It is modified.

*Page 8 and after: no need to bold "official run" and "test run". I find this distracting in the rest of the document.*

**Reply**: Thanks. They are modified.

*Page 12, line 25: need units after 0.4 (should read thinner than 0.4 m)*

**Reply**: Thanks. It is corrected.

*Page 12, line 32: Yang reference is (2015), but in bibliography it is 2014. Please check on the year of this publication.*

**Reply**: Thanks. This mistake is corrected.

*Figure 1: the words of the regions are hard to see (especially Kara and Beaufort). I suggest putting a white background for these words.*

**Reply**: Thanks for this suggestion. It is modified.

*Figure 3: Don't need the word "resp" when doing comparisons. Usually when compar- isons are done in this manner you just state the contrasting item. The green (red) line represents the mean bias for March (November) of each year.*

**Reply**: Thanks. They are deleted.

*Figure 4: Bottom row. It is hard to see the orange line. I suggest choosing a separate color not in the colorbar. Same comment about "resp" as above. Say Top Row, Middle Row, Bottom Row.*

**Reply**: Thanks. The suggestions are taken in the revision.

*Figure 5: The vertical axis changes on each plot. Please plot each item with the same vertical axis.*

**Reply**: Thanks. The figure is modified as suggested.

*Figures 6, 7, 8: same comment about "resp" and bold test run, official run.*

**Reply**: Thanks. They are modified as suggested.

***Reviewer #2:***

*(1) Page 2, line 6: "winter season" should be "cold season".*

**Reply**: We agree with the reviewer, and also change throughout the manuscript.

*(2) Page 2, line 13, the full name of TOPAZ should be given in the Abstract.*

**Reply**: The origin of the name TOPAZ is from a European project ***(Towards) an Operational Prediction system for the North Atlantic European coastal Zones.*** However time has passed and we now consider TOPAZ as a brand name and no longer as an acronym.

*(3) Page 2, line 20, should "contents" be corrected to "contains" ?*

**Reply**: Thanks. It is replaced by "contains".

*(4) Page 2, line 28, the "Keywords" should be revised, e.g., a lot of readers do not know "OSE" and "DFS".*

**Reply**: These words are replaced by "Observing System Experiment" and "Degrees of Freedom for Signal".

*(5) Page 3, line 22, "draft" should be "freeboard".*

**Reply**: Thanks. This mistake is corrected by freeboard.

*(6) Page 4, line 13: Yang et al. (2015) should be (2014).*

**Reply**: Thanks. The reference is corrected by Yang et al. (2014).

*(7) Page 4, line 14: "LSEIK" should be defined.*

**Reply**: It is defined by with the Localized Singular Evolutive Interpolated Kalman filter (LSEIK, *ref*. Nerger et al., 2005).

*(8) Page 4, line 26, "Xie et al., 2016" is frequently referred in this MS, this should be corrected, as it has not been accepted, the authors even have not tell us the journal they submitted to.*

**Reply**: Xie et al. (2016) was submitted to *Ocean Science discussion* at the moment, and is available online as doi:10.5194/os-2016-38. http://www.ocean-sci-discuss.net/os-2016-38/

*(9) Page 7, line 15, is "TOPAZ equivalent ice thickness" "TOPAZ model mean ice thick- ness"?*

**Reply**: This statement is modified as "The TOPAZ ice thicknesses shown in Fig.2 are at the same locations and times as the observations."

*(10) Page 7, line 17, "RMSD" is not defined here.*

**Reply**: Thanks. This definition is added.

*(11) Page 7, line 29, you only assimilate the SMOS data less than 0.40 m, why not 0.50 m? As you referred, "the penetration depth into sea ice is about 0.5 m". Although you mentioned that "the effect of ice melting may lead to a saturation thickness of less than 0.4 m", but for this paper, you run the experiments in the cold season, basically there is no melting in the sea ice surface. If you increase the upper limit, more SMOS observation data is available, thus stronger influence/correction to the TOPAZ system is expected. In Yang et al. (2014), they use an upper limit of 1.0 m.*

**Reply**: It is correct that by raising the threshold to 1 m we would increase the influence of the observation. However the observation error becomes very large above 0.4 m, so we do not expect that we are loosing much information (see also Fig. A in the answer to Reviewer #1). The main motivation for rejecting the observation above this threshold is that there is an obvious bias between model and observation beyond this threshold. Data assimilation with bias is problematic because the correction of the bias may be transferred to other variables via the multivariate updates of the scheme. We have therefore taken a cautious approach and decided not to use the data > 0.4 m for the moment. The word "multivariate" is added on p. 8, l. 8

*(12) Page 8, line 3, 4, 7: "thick" should be "thickness"?*

**Reply**: Here, it means the sea-ice thickness simulated by the model is too thick relative to the SMOS-Ice data. We want to keep the indication of the sign of the bias (too thick instead of too thin).

*(13) Page 8, line 19, $y_{smos}$ is not defined.*

**Reply**: Thanks. The definition is added in the revision.

*(14) Page 9, line 10: "SMOS-Ice" is forgotten in Table 1.*

**Reply**: Table 1 lists the observations assimilated in the present TOPAZ system. This is clarified in the revision.

*(15) Page 11, line 1: In the Beaufort Sea, there are some sea ice draft measure- ments from Beaufort Gyre Exploration Project (BGEP) by upward-looking sonar (ULS) moorings located in the Beaufort Sea (http://www.whoi.edu/beaufortgyre). Also, there are some sea ice thickness data obtained from autonomous ice mass balance (IMB; http://imb.erdc.dren.mil ). I would suggest the authors to use these data as the inde- pendent ice thickness observations in the evaluation of their model results.*

**Reply**: Thanks. The two buoys from the IMB have been used to validate the sea ice thickness as the Fig. 7 in the revision. As the buoys are far away from assimilated observation, the impact is small. Still there is a slight improvement.

*(16) Page 12, line 32, an "a" is missing before "slight". (17) Page 13, line 24, should be "In addition".*

**Reply**: Thanks. It is added in the revision.

---

## Author Response (AR2)

**Response to Editor's comments**

We thank the editor for her detailed comments. The major changes in the revision have been addressed by the following change:

- Added more validations compared to moorings from BGEP and two buoys from IMB in March 2014.
- Add one figure about the observation uncertainty of SMOS-Ice
- Highlight the novelty of the present study.
- Improve the clarity of the paper by revising the text.
- Add more informations about the validation period and data sets of sea ice thickness in the TOPAZ reanalysis.

In the following we provide a detailed answer to the editor comment. Her comments are repeated in black and our answers are given in red.

*This paper is not ground breaking, in that it demonstrates that assimilating ice thickness into a coupled ice-ocean model does exactly what you would expect it to do, move the ice thickness estimate towards observations. A very similar paper has already been published in JGR-Ocean (Yang et al. 2014), documenting the impact of assimilating SMOS thicknesses < 1m into the MITgcm with a Kalman Filtering method. Yang et al. has more extensive validation with independent ice thickness measurements than you perform in this paper, with data sets that are available to you. It appears your results are similar to Yang et al. (2014), and I wonder if any increase in impact of SMOS is actually because your original run is actually simulating ice thickness worse than the model used in Yang et al. (2014). Given the fact that this is a reproduction of a previous study, with a similar assimilation method (some differences that are worth note) and different model, your results do not present much additional information to the reader. That said,this result is useful to the community of future sea ice forecasters and analysts who may use the TOPAZ model, which is providing medium-range forecasts unlike the seasonal forecasts of Yang et al. (2014). So I am inclined to reconsider the paper after revision.*

It is correct that assimilation of ice thickness is expected to bring the model closer to observations. However, it is not warranted that improvements in one location do not lead to degradations in other locations. Also the multivariate properties of our strongly coupled data assimilation method do not degrade the ocean variables (SST, SSH) and even lead to some slight benefit for ice

concentration. We have noticed some changes in the ice drift and surface currents but there were no observations able to indicate whether these are changes for the better. It is correct that the current manuscript has similarities with the one from Yang et al. (2014); that the benefit from assimilating SMOS-Ice is not as large and as expected. However, *two is more convincing than one* and we think that some additional finding makes it useful to the user community. The possibility that our results start from a much worse ice thickness than Yang et al. (2014) does not seem to hold from the results at hand: although we cannot compare the models in different periods, their thickness offset is reported about 1.5 m against BGEP 2011 and IMB 2011 data, whereas our thickness offset is between 10 cm and 80 m between our results and IMB (Fig. 8 in the revised manuscript).

We have tried to better present in the new manuscript the novelty in our study compared to Yang et al. (2014) namely: 1) Assimilation is validated from the beginning to the end of the cold period while the experimental period in Yang et al. (2014) is only from November 2011 to January 2012. 2) The conclusion from Yang et al. (2014) still holds when using a much more extensive observation network (they only assimilate sea ice concentration and SMOS-Ice while we assimilate T-S profiles, altimetry, SST, sea ice drift in addition. 3) We have verified that we do not degrade the performance in the ocean 4) There is a quantification of the relative impact of SMOS-Ice with respect to a full observational network. 5) We present and validate changes to the European monitoring service (Arctic MFC Copernicus). 6) It is always good to show that conclusions from a paper can be verified with a different system (larger ensemble size, different model, slightly different assimilation method, different implementation and different observation network).

*I welcome the addition of independent thickness observations for verification, however you did not present as much verification data as is available to you. And as you point out in your response to the reviewer, the location of buoys compared against is upstream from where data is assimilated. I also find that the paper can still benefit from attention to the error characteristics of the thickness data. This is very important for understanding R in the Kalman gain. Figure A in the response to reviewers was helpful, and you could perhaps describe these error characteristics in the manuscript.*

**Reply**: Thanks. We have now extended the validation with the other independent measurement data sets available to us. The new validation supports the previous conclusions. We have added the figure about observation error of SMOS-Ice.

*Please provide more information regarding the validation data set, used to assess skill in estimating ice thickness. Xie et al. (2016) also insufficiently describes the ice thickness data. You should describe the data density and how it varies spatially and temporally over the full time period. This paper in Ocean Science Discussions does not provide a discussion of the measurement errors, which can be substantial. Without this discussion it is impossible to assess if biases are in the model or observations (as you correctly point out in your manuscript). It is for this reason that I suggest you consider using more ice thickness data to see if you can tease this information out. However, I do understand that you may find there is limited data of thin ice thickness with which to directly verify the SMOS data set and model biases.*

**Reply**: Thanks. We have improved the presentation of the data set used for validation in our manuscript.

*Only two buoy trajectories were used in this paper for validation. I assume that this is because these are the only IMB data that overlaps with your time period for model runs. There are many other data sources you can use for independent verification. Ice Bridge will be useful for March in the high Arctic, though I suspect you are not assimilating SMOS data here. There are year long moorings in the Beaufort Sea with ULS. The data, described in Krishfield et al. (2014), for these is freely available from WHOI at http://www.whoi.edu/page.do?pid=66559. Note that Krishfield processed this data, and had to filter out wave action in summer months as the Beaufort has become Marginal Ice Zone. The data processing may imply a minimum ice thickness that can be resolved, and you should check this.*

**Reply**: Thank you for the recommendations. We have now extended the validation with the recommended observational data sets. We have made a note of the processing applied to the mooring data and the IceBridge data is indeed always thicker than 1m. The validation against the *Ice Bridge* data was performed (see Figure below) but is not more conclusive that the other data sets: only a handful of observation points are affected by the assimilation of SMOS data. We have thus mentioned that validation against IceBridge was performed but that it is not presented because we get similar results.

[Figure]

Comparison of the SITs in the two assimilation runs with the averaged SITs of IceBridge Quick Look from the National Snow and Ice Data Centre in March 2014. (a) Locations of the observed SIT with their standard deviation (unit: m); (b) Scatter of the observed thickness and the simulated thickness of Official (Test) Run shown as the blue (pink) color. The blue (pink) line represents the regression lines. (c) Distribution of the sea ice thickness differences between the Official Run and the observed. (d) Distribution of the sea ice thickness difference between the two assimilation runs.

*I am aware of other moorings deployed in the Beaufort Sea, in the seasonal ice zone. Consider contacting Humfrey Melling regarding this data, though it may be propriety. Shell had moorings in the Chukchi Sea and they have been providing data freely to researchers. This is information that is for your interest, as I am sure you Consider the Fram Strait moorings too, though these are not recording thin ice. Some, but not all, of these data are included in the Lindsay et al. (2013) unified sea-ice thickness product.*

**Reply**: Thanks. Most of the data sets do not provide SIT, or stopped after 2012, or are only in summer time. Although validation of ice thickness is

important, validation of sea levels, SST, ice concentrations are equally important and just as independent.

*Please expand TOPAZ where it is first introduced. Not everyone will be familiar with the model, and the brand name is not in common usage.*

**Reply**: The Acronym TOPAZ, originates from an European project ("Towards an Operational Prediction System for the North Atlantic European coastal Zones"). Since 2004, our focus has gradually evolved to the Arctic and the acronym has actually become a brand name, known as such in the ocean data assimilation community. None of the publications about TOPAZ system mention the meaning of the acronym any more and we would find it confusing to remind it here.

*page 2, line 5: There are more recent references to the decrease in sea ice extent. Also, Shimada et al. is a discussion of the possible implications of reduced sea ice extent and not a presentation of the observation of reduced extent.*

**Reply**: Thanks. The related reference are replaced by "Comiso et al., 2008; Stroeve et al., 2012".

*page 5, line 4: In the one catagory model is there a lower limit on ice thickness that is not considered open water. For example some older two level models consider ice less than 0.5m to be in the open water catagory. Please clarify, as this is exceptionally important for your assimilation scheme.*

**Reply**: Thanks for this important point. In the model, the thickness of sea ice as a minimal limitation thickness of 0.1m. This threshold is relatively low in the community and is not limiting the use of SMOS since we use the ensemble mean of a 100 members ensemble, which can take values down to 1 mm. We mention it during the model description and its implication during the validation.

*line 14-15: "which amplitude is" -> "with amplitude"*
**Reply**: It is corrected.

*line 27: bracket -> brackets*
**Reply**: It is corrected.

*line 31: missing f superscript from P*
**Reply**: Thanks. The related illustration has been changed.

*line 33: anomaly -> anomalies*
**Reply**: It is corrected.

*page 6, line 1: forecastd - > forecast*
**Reply**: It is corrected.

*Page 6: Please check you have defined all your variables.*
*line 10: put symbol for ensemble mean just after when you introduce it. So it is clear the equation is calculating this. Ditto for ensemble anomaly in line 12.*
**Reply**: Thanks. The order has been changed again.

*Equation 4: This is actually the mean difference between model and observation. While you are correct that this includes both the observation and model bias, I do not find Bias to be the most intuitive label for this quantity. However I am willing to concede provided it is very clear to the reader through out that the bias is not the model or observation bias.*
**Reply**: The bias is the expected value of the difference between the model and the truth. Here the truth is unknown, and the bias is the sum of the model and observation bias. Labeling this quantity as bias is common in the data assimilation community as model bias is often much larger than observation bias. We have tried to clarify that in the new version of the manuscript.

*page 8, line 13: Sentence incomplete*
**Reply**: Thank you. It is corrected.

*page 9, line 1: You need to describe the uncertainties. Including their magnitude, any variance and periodicity in this.*
**Reply**: Thanks. More information about the observation uncertainties are added in Page 9 lines: 15- 34.

*line 2: As you are disregarding all data more that 0.4m, is it not irrelevant that the upper, saturation, limit on SMOS observations is 5m. I see that you refer to this point later in the manuscript, where you point out there is very little SMOS data assimilated in the Beaufort in March. You could clarify these points when you expand the information about the SMOS uncertainty.*
**Reply**: The upper uncertainty of SIT from SMOS is set to 5 m (variance of 25 $m^2$). When the saturation threshold of observation uncertainty is reached we

reject the observation whatever the value of SIT. However, it is very seldom that SMOS-Ice is less than 0.4 m with an observation error of more than 25 m^2 (see Fig 4 in the revised manuscript), so we do not think it has any influence on our results.

*line 8: "within the beginnings" might read better as "at the onset"*
**Reply**: Thank you, it is replaced in the revision.

*page 11, line 11: "we are validating" -> "we validate"*
**Reply**: Thanks. It is changed.

*page 13, line 10: Remove "the ice tethered profiles (ITP), which are"*
**Reply**: Thank you, it is corrected as the suggestion.

*line 13: _[Dimpacts -> impact*
**Reply**: Thanks. It is corrected.

*page 14, line 18: thick -> thickness*
**Reply**: It is replaced by thicker.

*line 27: consistently -> consistent*
**Reply**: Thanks. It is corrected.

*Fig. 5: Expand acronyms in titles. You do not explain what hice, icec is etc.*
**Reply**: Thanks. We have tried to limit the use of acronym.

*Fig. 6, caption: sea-ice -> sea ice*
**Reply**: Thanks. It is corrected.

**Response to Editor's comments**

We thank the editor for her detailed comments. The major changes in the revision have been addressed by the following change:

- Added more validations compared to moorings from BGEP, and two buoys from IMB in March 2014.
- Add one figure about the observation uncertainty of SMOS-Ice.
- Highlight the novelty of the present study.
- Improve the clarity of the paper by revising the text.
- Add more informations about the validation period and data sets of sea ice thickness in the TOPAZ reanalysis.

In the following we provide a detailed answer to the editor comment. Her comments are repeated in black and our answers are given in red.

*This paper is not ground breaking, in that it demonstrates that assimilating ice thickness into a coupled ice-ocean model does exactly what you would expect it to do, move the ice thickness estimate towards observations. A very similar paper has already been published in JGR-Ocean (Yang et al. 2014), documenting the impact of assimilating SMOS thicknesses < 1m into the MITgcm with a Kalman Filtering method. Yang et al. has more extensive validation with independent ice thickness measurements than you perform in this paper, with data sets that are available to you. It appears your results are similar to Yang et al. (2014), and I wonder if any increase in impact of SMOS is actually because your original run is actually simulating ice thickness worse than the model used in Yang et al. (2014). Given the fact that this is a reproduction of a previous study, with a similar assimilation method (some differences that are worth note) and different model, your results do not present much additional information to the reader. That said,this result is useful to the community of future sea ice forecasters and analysts who may use the TOPAZ model, which is providing medium-range forecasts unlike the seasonal forecasts of Yang et al. (2014). So I am inclined to reconsider the paper after revision.*

It is correct that assimilation of ice thickness is expected to bring the model closer to observations. However, it is not warranted that improvements in one location do not lead to degradations in other locations. Also the multivariate properties of our strongly coupled data assimilation method do not degrade the ocean variables (SST, SSH), and even lead to some slight benefit for ice

concentration. We have noticed some changes in the ice drift and surface currents but there were no observations able to indicate whether these are changes for the better. It is correct that the current manuscript has similarities with the one from Yang et al. (2014); that the benefit from assimilating SMOS-Ice is not as large and as expected. However, two is more convincing than one and we think that some additional finding makes it useful to the user community. The possibility that our results start from a much worse ice thickness than Yang et al. (2014) does not seem to hold from the results at hand: although we cannot compare the models in different periods, their thickness offset is reported about 1.5 m against BGEP 2011 and IMB 2011 data, whereas our thickness offset is between 10 cm and 80 m between our results and IMB (Fig. 8 in the revised manuscript).

We have tried to better present in the new manuscript the novelty in our study compared to Yang et al. (2014) namely: 1) Assimilation is validated from the beginning to the end of the cold period while the experimental period in Yang et al. (2014) is only from November 2011 to January 2012, 2) The conclusion from Yang et al. (2014) still holds when using a much more extensive observation network (they only assimilate sea ice concentration and SMOS-Ice while we assimilate T-S profiles, altimetry, SST, sea ice drift in addition. 3) We have verified that we do not degrade the performance in the ocean 4) There is a quantification of the relative impact of SMOS-Ice with respect to a full observational network. 5) We present and validate changes to the European monitoring service (Arctic MFC Copernicus). 6) It is always good to show that conclusions from a paper can be verified with a different system (larger ensemble size, different model, slightly different assimilation method, different implementation and different observation network).

*I welcome the addition of independent thickness observations for verification, however you did not present as much verification data as is available to you. And as you point out in your response to the reviewer, the location of buoys compared against is upstream from where data is assimilated. I also find that the paper can still benefit from attention to the error characteristics of the thickness data. This is very important for understanding R in the Kalman gain. Figure A in the response to reviewers was helpful, and you could perhaps describe these error characteristics in the manuscript.*

Jiping et al. Mac 17/10/2016 10:33
Formatted ... [8]
Jiping et al. Mac 17/10/2016 10:33
Jiping et al. Mac 17/10/2016 10:33
Jiping et al. Mac 17/10/2016 10:33
Formatted ... [10]
Jiping et al. Mac 17/10/2016 10:33
Formatted ... [11]
Jiping et al. Mac 17/10/2016 10:33
Jiping et al. Mac 17/10/2016 10:33
Jiping et al. Mac 17/10/2016 10:33
Formatted ... [13]
Jiping et al. Mac 17/10/2016 10:33
Jiping et al. Mac 17/10/2016 10:33
Formatted ... [12]
Jiping et al. Mac 17/10/2016 10:33
Formatted ... [14]
Jiping et al. Mac 17/10/2016 10:33
Formatted ... [15]
Jiping et al. Mac 17/10/2016 10:33
Formatted ... [16]
Jiping et al. Mac 17/10/2016 10:33
Jiping et al. Mac 17/10/2016 10:33
Formatted ... [17]
Jiping et al. Mac 17/10/2016 10:33
Jiping et al. Mac 17/10/2016 10:33
Formatted ... [18]
Jiping et al. Mac 17/10/2016 10:33
Jiping et al. Mac 17/10/2016 10:33
Formatted ... [19]
Jiping et al. Mac 17/10/2016 10:33
Jiping et al. Mac 17/10/2016 10:33
Formatted ... [20]
Jiping et al. Mac 17/10/2016 10:33
Jiping et al. Mac 17/10/2016 10:33
Formatted ... [21]
Jiping et al. Mac 17/10/2016 10:33
Jiping et al. Mac 17/10/2016 10:33
Formatted ... [22]
Jiping et al. Mac 17/10/2016 10:33
Jiping et al. Mac 17/10/2016 10:33
Formatted ... [23]
Jiping et al. Mac 17/10/2016 10:33
Jiping et al. Mac 17/10/2016 10:33
Formatted ... [24]
Jiping et al. Mac 17/10/2016 10:33
Jiping et al. Mac 17/10/2016 10:33
Formatted ... [25]
Jiping et al. Mac 17/10/2016 10:33
... [26]
Jiping et al. Mac 17/10/2016 10:33
Formatted ... [27]
Jiping et al. Mac 17/10/2016 10:33
Jiping et al. Mac 17/10/2016 10:33
Formatted ... [28]
Jiping et al. Mac 17/10/2016 10:33
Jiping et al. Mac 17/10/2016 10:33
Formatted ... [29]
Jiping et al. Mac 17/10/2016 10:33

**Reply**: Thanks. We have now extended the validation with the other independent measurement data sets available to us. The new validation supports the previous conclusions. We have added the figure about observation error of SMOS-Ice.

*Please provide more information regarding the validation data set, used to assess skill in estimating ice thickness. Xie et al. (2016) also insufficiently describes the ice thickness data. You should describe the data density and how it varies spatially and temporally over the full time period. This paper in Ocean Science Discussions does not provide a discussion of the measurement errors, which can be substantial. Without this discussion it is impossible to assess if biases are in the model or observations (as you correctly point out in your manuscript). It is for this reason that I suggest you consider using more ice thickness data to see if you can tease this information out. However, I do understand that you may find there is limited data of thin ice thickness with which to directly verify the SMOS data set and model biases.*

**Reply**: Thanks. We have improved the presentation of the data set used for validation in our manuscript.

*Only two buoy trajectories were used in this paper for validation. I assume that this is because these are the only IMB data that overlaps with your time period for model runs. There are many other data sources you can use for independent verification. Ice Bridge will be useful for March in the high Arctic, though I suspect you are not assimilating SMOS data here. There are year long moorings in the Beaufort Sea with ULS. The data, described in Krishfield et al. (2014), for these is freely available from WHOI at http://www.whoi.edu/page.do?pid=66559. Note that Krishfield processed this data, and had to filter out wave action in summer months as the Beaufort has become Marginal Ice Zone. The data processing may imply a minimum ice thickness that can be resolved, and you should check this.*

**Reply**: Thank you for the recommendations. We have now extended the validation with the recommended observational data sets. We have made a note of the processing applied to the mooring data and the IceBridge data is indeed always thicker than 1m. The validation against the *Ice Bridge* data was performed (see Figure below) but is not more conclusive that the other data sets: only a handful of observation points are affected by the assimilation of SMOS data. We have thus mentioned that validation against IceBridge was performed but that it is not presented because we get similar results.

Jiping et al. Mac 17/10/2016 10:33

Jiping et al. Mac 17/10/2016 10:33

Jiping et al. Mac 17/10/2016 10:33

Jiping et al. Mac 17/10/2016 10:33

Jiping et al. Mac 17/10/2016 10:33

Jiping et al. Mac 17/10/2016 10:33

Jiping et al. Mac 17/10/2016 10:33

Jiping et al. Mac 17/10/2016 10:33

Jiping et al. Mac 17/10/2016 10:33

Jiping et al. Mac 17/10/2016 10:33

Jiping et al. Mac 17/10/2016 10:33

Jiping et al. Mac 17/10/2016 10:33

Jiping et al. Mac 17/10/2016 10:33

Jiping et al. Mac 17/10/2016 10:33

Jiping et al. Mac 17/10/2016 10:33

Jiping et al. Mac 17/10/2016 10:33

Jiping et al. Mac 17/10/2016 10:33

Jiping et al. Mac 17/10/2016 10:33

Jiping et al. Mac 17/10/2016 10:33

Jiping et al. Mac 17/10/2016 10:33

Jiping et al. Mac 17/10/2016 10:33

Jiping et al. Mac 17/10/2016 10:33

[Figure]

Comparison of the SITs in the two assimilation runs with the averaged SITs of IceBridge Quick Look from the National Snow and Ice Data Centre in March 2014. (a) Locations of the observed SIT with their standard deviation (unit: m); (b) Scatter of the observed thickness and the simulated thickness of Official (Test) Run shown as the blue (pink) color. The blue (pink) line represents the regression lines. (c) Distribution of the sea ice thickness differences between the Official Run and the observed. (d) Distribution of the sea ice thickness difference between the two assimilation runs.

*I am aware of other moorings deployed in the Beaufort Sea, in the seasonal ice zone. Consider contacting Humfrey Melling regarding this data, though it may be propriety. Shell had moorings in the Chukchi Sea and they have been providing data freely to researchers. This is information that is for your interest, as I am sure you Consider the Fram Strait moorings too, though these are not recording thin ice. Some, but not all, of these data are included in the Lindsay et al. (2013) unified sea-ice thickness product.*

**Reply**: Thanks. Most of the data sets do not provide SIT, or stopped after 2012, or are only in summer time. Although validation of ice thickness is

Jiping et al. Mac 17/10/2016 10:33

Jiping et al. Mac 17/10/2016 10:33

Jiping et al. Mac 17/10/2016 10:33

Jiping et al. Mac 17/10/2016 10:33

Jiping et al. Mac 17/10/2016 10:33

Jiping et al. Mac 17/10/2016 10:33

Jiping et al. Mac 17/10/2016 10:33

Jiping et al. Mac 17/10/2016 10:33

Jiping et al. Mac 17/10/2016 10:33

important, validation of sea levels, SST, ice concentrations are equally important and just as independent.

*Please expand TOPAZ where it is first introduced. Not everyone will be familiar with the model, and the brand name is not in common usage.*

**Reply**: The Acronym TOPAZ, originates from an European project ("Towards an Operational Prediction System for the North Atlantic European coastal Zones"). Since 2004, our focus has gradually evolved to the Arctic and the acronym has actually become a brand name, known as such in the ocean data assimilation community. None of the publications about TOPAZ system mention the meaning of the acronym any more and we would find it confusing to remind it here.

*page 2, line 5: There are more recent references to the decrease in sea ice extent. Also, Shimada et al. is a discussion of the possible implications of reduced sea ice extent and not a presentation of the observation of reduced extent.*

**Reply**: Thanks. The related reference are replaced by "Comiso et al., 2008; Stroeve et al., 2012".

*page 5, line 4: In the one catagory model is there a lower limit on ice thickness that is not considered open water. For example some older two level models consider ice less than 0.5m to be in the open water catagory. Please clarify, as this is exceptionally important for your assimilation scheme.*

**Reply**: Thanks for this important point. In the model, the thickness of sea ice as a minimal limitation thickness of 0.1m. This threshold is relatively low in the community and is not limiting the use of SMOS since we use the ensemble mean of a 100 members ensemble, which can take values down to 1 mm. We mention it during the model description and its implication during the validation.

*line 14-15: "which amplitude is" -> "with amplitude"*
**Reply**: It is corrected.

*line 27: bracket -> brackets*
**Reply**: It is corrected.

Jiping et al. Mac 17/10/2016 10:33

Jiping et al. Mac 17/10/2016 10:33

Jiping et al. Mac 17/10/2016 10:33

Jiping et al. Mac 17/10/2016 10:33

Jiping et al. Mac 17/10/2016 10:33

Jiping et al. Mac 17/10/2016 10:33

Jiping et al. Mac 17/10/2016 10:33

Jiping et al. Mac 17/10/2016 10:33

Jiping et al. Mac 17/10/2016 10:33

Jiping et al. Mac 17/10/2016 10:33
**Formatted** ... [34]

Jiping et al. Mac 17/10/2016 10:33

Jiping et al. Mac 17/10/2016 10:33
**Formatted** ... [35]

Jiping et al. Mac 17/10/2016 10:33
**Formatted** ... [36]

Jiping et al. Mac 17/10/2016 10:33
**Formatted** ... [37]

Jiping et al. Mac 17/10/2016 10:33

Jiping et al. Mac 17/10/2016 10:33
**Formatted** ... [38]

Jiping et al. Mac 17/10/2016 10:33
**Formatted** ... [39]

Jiping et al. Mac 17/10/2016 10:33
**Formatted** ... [40]

Jiping et al. Mac 17/10/2016 10:33

Jiping et al. Mac 17/10/2016 10:33
**Formatted** ... [41]

Jiping et al. Mac 17/10/2016 10:33
**Formatted** ... [42]

Jiping et al. Mac 17/10/2016 10:33
**Formatted** ... [43]

Jiping et al. Mac 17/10/2016 10:33
**Formatted** ... [44]

Jiping et al. Mac 17/10/2016 10:33

Jiping et al. Mac 17/10/2016 10:33
**Formatted** ... [45]

Jiping et al. Mac 17/10/2016 10:33
**Formatted** ... [46]

*line 31: missing f superscript from P*
**Reply**: Thanks. The related illustration has been changed.

*line 33: anomaly -> anomalies*
**Reply**: It is corrected.

*page 6, line 1: forecastd - > forecast*
**Reply**: It is corrected.

*Page 6: Please check you have defined all your variables.*
*line 10: put symbol for ensemble mean just after when you introduce it. So it is clear the equation is calculating this. Ditto for ensemble anomaly in line 12.*
**Reply**: Thanks. The order has been changed again.

*Equation 4: This is actually the mean difference between model and observation. While you are correct that this includes both the observation and model bias, I do not find Bias to be the most intuitive label for this quantity. However I am willing to concede provided it is very clear to the reader through out that the bias is not the model or observation bias.*
**Reply**: The bias is the expected value of the difference between the model and the truth. Here the truth is unknown, and the bias is the sum of the model and observation bias. Labeling this quantity as bias is common in the data assimilation community as model bias is often much larger than observation bias. We have tried to clarify that in the new version of the manuscript.

*page 8, line 13: Sentence incomplete*
**Reply**: Thank you. It is corrected.

*page 9, line 1: You need to describe the uncertainties. Including their magnitude, any variance and periodicity in this.*
**Reply**: Thanks. More information about the observation uncertainties are added in Page 9 lines: 15- 34.

*line 2: As you are disregarding all data more that 0.4m, is it not irrelevant that the upper, saturation, limit on SMOS observations is 5m. I see that you refer to this point later in the manuscript, where you point out there is very little SMOS data assimilated in the Beaufort in March. You could clarify these points when you expand the information about the SMOS uncertainty.*
**Reply**: The upper uncertainty of SIT from SMOS is set to 5 m (variance of 25 $m^2$). When the saturation threshold of observation uncertainty is reached we

Jiping et al. Mac 17/10/2016 10:33
**Formatted** ... [47]

Jiping et al. Mac 17/10/2016 10:33
**Formatted** ... [48]

Jiping et al. Mac 17/10/2016 10:33
**Formatted** ... [49]

Jiping et al. Mac 17/10/2016 10:33
**Formatted** ... [50]

Jiping et al. Mac 17/10/2016 10:33
**Formatted** ... [51]

Jiping et al. Mac 17/10/2016 10:33
**Formatted** ... [52]

Jiping et al. Mac 17/10/2016 10:33

Jiping et al. Mac 17/10/2016 10:33
**Formatted** ... [54]

Jiping et al. Mac 17/10/2016 10:33

Jiping et al. Mac 17/10/2016 10:33
**Formatted** ... [53]

Jiping et al. Mac 17/10/2016 10:33
**Formatted** ... [55]

Jiping et al. Mac 17/10/2016 10:33
**Formatted** ... [56]

Jiping et al. Mac 17/10/2016 10:33
**Formatted** ... [57]

Jiping et al. Mac 17/10/2016 10:33
**Formatted** ... [58]

Jiping et al. Mac 17/10/2016 10:33

Jiping et al. Mac 17/10/2016 10:33
**Formatted** ... [59]

Jiping et al. Mac 17/10/2016 10:33

Jiping et al. Mac 17/10/2016 10:33
**Formatted** ... [60]

Jiping et al. Mac 17/10/2016 10:33
**Formatted** ... [61]

Jiping et al. Mac 17/10/2016 10:33
**Formatted** ... [62]

Jiping et al. Mac 17/10/2016 10:33

Jiping et al. Mac 17/10/2016 10:33
**Formatted** ... [63]

Jiping et al. Mac 17/10/2016 10:33

Jiping et al. Mac 17/10/2016 10:33
**Formatted** ... [64]

Jiping et al. Mac 17/10/2016 10:33

Jiping et al. Mac 17/10/2016 10:33
**Formatted** ... [66]

Jiping et al. Mac 17/10/2016 10:33

reject the observation whatever the value of SIT. However, it is very seldom that SMOS-Ice is less than 0.4 m with an observation error of more than 25 m^2 (see Fig 4 in the revised manuscript), so we do not think it has any influence on our results.

*line 8: "within the beginnings" might read better as "at the onset"*
**Reply**: Thank you, it is replaced in the revision.

*page 11, line 11: "we are validating" -> "we validate"*
**Reply**: Thanks. It is changed.

*page 13, line 10: Remove "the ice tethered profiles (ITP), which are"*
**Reply**: Thank you, it is corrected as the suggestion.

*line 13: _[Dimpacts -> impact*
**Reply**: Thanks. It is corrected.

*page 14, line 18: thick -> thickness*
**Reply**: It is replaced by thicker.

*line 27: consistently -> consistent*
**Reply**: Thanks. It is corrected.

*Fig. 5: Expand acronyms in titles. You do not explain what hice, icec is etc.*
**Reply**: Thanks. We have tried to limit the use of acronym.

*Fig. 6, caption: sea-ice -> sea ice*
**Reply**: Thanks. It is corrected.

Jiping et al. Mac 17/10/2016 10:33

Jiping et al. Mac 17/10/2016 10:33

Jiping et al. Mac 17/10/2016 10:33

Jiping et al. Mac 17/10/2016 10:33

Jiping et al. Mac 17/10/2016 10:33

Jiping et al. Mac 17/10/2016 10:33

Jiping et al. Mac 17/10/2016 10:33

Jiping et al. Mac 17/10/2016 10:33

Jiping et al. Mac 17/10/2016 10:33

Jiping et al. Mac 17/10/2016 10:33

Jiping et al. Mac 17/10/2016 10:33

Jiping et al. Mac 17/10/2016 10:33

Jiping et al. Mac 17/10/2016 10:33

Jiping et al. Mac 17/10/2016 10:33

Jiping et al. Mac 17/10/2016 10:33

[revised manuscript text omitted]

$$\bar{\mathbf{x}}^{\mathbf{a}} = \bar{\mathbf{x}}^{\mathbf{f}} + \mathbf{K}(\mathbf{y} - \mathbf{H}\bar{\mathbf{x}}^{\mathbf{f}}),$$

The analyzed ensemble anomaly is calculated as follows:

$$\mathbf{A}^{\mathbf{a}} = \mathbf{A}^{\mathbf{f}} - \frac{1}{2}\mathbf{KHA}^{\mathbf{f}}.$$

The full ensemble is reconstructed by adding the two terms as follows:

$$\mathbf{X}^{a} = \mathbf{A}^{a} + \overline{\mathbf{x}^{a}}\mathbf{I}_{N} \qquad\qquad (3),$$

where $\mathbf{X}^{a}$ is the matrix of the updated model states after assimilation.

An overview of the observations assimilated in the present TOPAZ system is given in Table 1. Observations are quality-controlled and superobed (Sakov et al., 2012). TOPAZ assimilates the following data sets on a weekly basis: the gridded SST from the Operational Sea Surface Temperature and Sea Ice Analysis system (OSTIA, Donlon et al., 2012); sea ice concentration from the Ocean & Sea Ice Satellite Application Facility (OSISAF); along-track Sea Level Anomaly by Collecte Localisation Satellites (CLS); delayed-mode profiles of temperature and salinity from Ifremer, and the sea ice drift during the 3 days prior to the analysis from the CERSAT (Centre ERS d'Archivage et de Traitement) of IFREMER (French Research Institute for Exploitation of the Sea). All these standard measurements are retrieved from http://marine.copernicus.eu. The SLA data and the sea ice drift data are assimilated asynchronously (see Sakov et al., 2010).

**3.    Bias analyses for thin ice thickness**

The TOPAZ system has computed a reanalysis at daily frequency for ocean and sea ice variables. Its sea ice thickness has been validated against in situ data and satellite observations in Xie et al. (2016). Data assimilation assumes that the model and observations errors are unbiased. In this section, we investigate the bias by analyzing the thickness misfits for thin sea ice during five cold seasons from 2010 to 2014.

SMOS-Ice products (version 2.1) are available during the cold season (from 15th October to 15th April) at daily frequency from 2010 and up to near-real time. The data set is provided by University of Hamburg (Kaleschke et al., 2012; Kaleschke et al., 2013; https://icdc.zmaw.de/1/daten/cryosphere/l3c-smos-sit.html).

Here, the daily averaged SITs of TOPAZ are compared to the observations. The spatial or temporal bias and Root Mean Square Difference (RMSD) are calculated as follows:

$$\mathbf{Bias} = \frac{1}{n}\sum_{i=1}^{n}(\mathbf{H}\bar{\mathbf{x}}_i^{\mathrm{f}} - \mathbf{y}_i) \qquad (4)$$

$$\mathbf{RMSD} = \sqrt{\frac{1}{n}\sum_{i=1}^{n}(\mathbf{H}\bar{\mathbf{x}}_i^{\mathrm{f}} - \mathbf{y}_i)^2}\,, \qquad (5)$$

where $\bar{\mathbf{x}}_i^{\mathrm{f}}$ is compared to observations at similar time, $\mathbf{H}$ is the observation operator (see eq. 1), and $n$ is the number of available observations within the calculation period. Note that, we compare the TOPAZ SITs to imperfect observations, which contains error and may also be biased. As such, the bias as formulated in Eq. 4 refers to the difference between the model and observation biases calculated against an unknown truth. Still it is reasonable to assume that the bias in the observation is smaller than in the model and that the bias obtained with Eq.4 mainly accounts for model bias.

Figure 2 shows the simulated SIT from the TOPAZ reanalysis as conditional expectations with respect to SMOS-Ice data sorted into bins of 5 cm. Again, the SITs from TOPAZ in Fig.2 are selected at same locations and time of observations. Overall, the SIT in TOPAZ tends to be overestimated. The overestimation varies from month to month and with the amplitude of SIT (more pronounced for thick ice). For SIT lower than 0.4 m, the match between the observations and TOPAZ is relatively good through the cold season. There is no clear bias between October and December but a slight increasing thick bias from January-April. For SIT larger than 0.4 m, TOPAZ clearly overestimates SIT compared to observations during October and February-April, while it underestimates it in November. The penetration depth for the L-Band microwaves frequency into sea ice is about 0.5 m (Kaleschke et al., 2010; Huntemann et al., 2014), and the effect of ice melting may lead to a saturation of the SIT for values lower than 0.4 m (see Heygster et al. 2009). For these reasons, assimilation of SITs thicker than 0.4 m appears as problematic because the large bias from observations or models may be transferred to other variables (e.g. in the ocean) via the multivariate properties of our

Jiping et al. Mac 17/10/2016 10:16

Jiping et al. Mac 17/10/2016 10:16

Jiping et al. Mac 17/10/2016 10:16

Jiping et al. Mac 17/10/2016 10:16

Jiping et al. Mac 17/10/2016 10:16

Jiping et al. Mac 17/10/2016 10:16

Jiping et al. Mac 17/10/2016 10:16

data assimilation method (note that TOPAZ uses strongly coupled data assimilation between the ocean and sea-ice). In the following we will only assimilate the SIT observations less than 0.4 m.

We now investigate whether there is an interannual, seasonal and spatial variability in the bias of SIT. Figure 3 shows the yearly bias (as defined in Eq. 4) for SIT thinner than 0.4 m during the period 2010-2014. After 2011, the thick bias is increasing, reaching a maximum of 0.1 m in 2014. There is some seasonality in the bias, and the thick bias is larger in March than in November. There is a large spatial variability in the distribution of the bias (right panel of Fig. 3), with the bias being largest in the Beaufort Sea and in the Kara Sea. We therefore select the periods of March and November 2014 to set the assimilation system in the most difficult situations.

[revised manuscript text omitted]

Jiping et al. Mac 17/10/2016 10:16

Jiping et al. Mac 17/10/2016 10:16

Jiping et al. Mac 17/10/2016 10:16

Jiping et al. Mac 17/10/2016 10:16

Jiping et al. Mac 17/10/2016 10:16

Jiping et al. Mac 17/10/2016 10:16

Jiping et al. Mac 17/10/2016 10:16

Jiping et al. Mac 17/10/2016 10:16

Jiping et al. Mac 17/10/2016 10:16

Jiping et al. Mac 17/10/2016 10:16

Jiping et al. Mac 17/10/2016 10:16

Jiping et al. Mac 17/10/2016 10:16

Jiping et al. Mac 17/10/2016 10:16

Jiping et al. Mac 17/10/2016 10:16

Jiping et al. Mac 17/10/2016 10:16

Jiping et al. Mac 17/10/2016 10:16

March, and perform identically. In November 2014, the observed sea ice drafts are thinner (< 1 m). The sea ice drafts from the OSE runs are again overestimated in all three locations. The averaged draft difference in the two runs is about 1 cm at the two moorings 2014a and 2014b, and about 16 cm at the mooring 2014d that is located closest to locations where SMOS-ICE has been assimilated (see Fig.5). We have also compared the two OSE runs in March 2014 with the NASA IceBridge SIT Quick Look data set (QL) available from National Snow and Ice Data Center. The analysis leads to similar conclusions (not shown), which is that assimilation of SMOS-ICE only yields to improvements of SIT near the ice edge near location where SMOS-ICE is assimilated but do not yield degradation in other places.

**5.     Relative impact of the SIT from SMOS-Ice**

In this Section, the quantitative benefit of assimilating SMOS-Ice into the TOPAZ system is compared to other observations assimilated. To do so, we evaluate a performance metric calculated during the analysis, the Degree of Freedom for Signal (DFS), which is widely used for such purposes (Rodgers 2000; Cardinali et al. 2004). During the assimilation, one can calculate the DFS as follows:

[revised manuscript text omitted]

Unknown

Jiping et al.Mac 17/10/2016 10:13

Jiping et al.Mac 17/10/2016 10:13

Jiping et al.Mac 17/10/2016 10:13

Jiping et al.Mac 17/10/2016 10:13

Jiping et al.Mac 17/10/2016 10:13

Jiping et al.Mac 17/10/2016 10:13

[Figure]

**Fig**. 12. Same as Figure 11 for November 2014

Jiping et al.Mac 17/10/2016 10:13

Unknown

Unknown

Jiping et al.Mac 17/10/2016 10:13

Jiping et al.Mac 17/10/2016 10:13

Jiping et al.Mac 17/10/2016 10:13

Jiping et al.Mac 17/10/2016 10:13

Jiping et al.Mac 17/10/2016 10:13

Jiping et al.Mac 17/10/2016 10:13

Jiping et al.Mac 17/10/2016 10:13

---

## Author Response (AR3)

**Response to Editor's comments**

We would like to thank you one more time for your constructive comments that has lead to improved quality and clarity of our manuscript.
You comments are repeated in black and our response is in red.
You will find the related change with track change in the new document.

pg 3, line 30: Cryostat2 -> Cryosat2
**Reply**: Thanks. We have used CryoSat-2.

pg 4, line 31 apply a post-processing -> apply post-processing
**Reply**: Thanks. It is corrected pg 8, line 8: What is superobed?
**Reply**: This is now explained (P8,L8):
"Observations are quality-controlled and superobed (i.e. the process of combining observations falling within the same model grid cell) as in Sakov et al., 2012."

pg 9, line 17: at the same
**Reply**: Thanks. It is corrected pg 9, line 18: as observations
**Reply**: Thanks. It is corrected pg 9, line 26: microwaves -> microwave
**Reply**: Thanks. It is corrected pg 9, line 30 and previous sentence: do you mean assimilation is problematic for SITs thinner than 0.4m. This is what the previous sentence suggests during the melt season. Did I miss-understand your point in the previous sentence? Perhaps reword or clarify the text here.
**Reply**: Thank you this was a mistake. The sentence is corrected (P9,L30):
"and the effect of ice melting leads to saturation beyond 0.4 m (see Heygster et al. 2009)"

pg 10, line 19: consider -> used, and remove 'are used' from end of sentence
**Reply**: Thanks. This is corrected pg 10, line 29: a 100 ensemble members -> 100 ensemble members
**Reply**: Thanks. It is corrected pg 15, line 11: near the location
**Reply**: Thanks. It is corrected pg 16, line 14: impact of the j'th
**Reply**: Thanks. It is corrected pg 16, line 18: the March -> March
**Reply**: Thanks. It is corrected pg 16, line 21: The profiles -> Profiles
**Reply**: Thanks. It is corrected pg 16, line 22: Arctic -> the Arctic
**Reply**: Thanks. It is corrected pg 16, line 25: impacts -> impact
**Reply**: Thanks. It is corrected pg 17, line 2: Consider starting sentence 'The study in this paper' rather than 'This study'
**Reply**: Thanks. It is corrected pg 17, line 33: These studies follow from the first ...
**Reply**: Thanks. It is corrected pg 18, line 4: spun up
**Reply**: Thanks. It is corrected pg 18, line 6: at time when -> the time period
**Reply**: Thanks. It is corrected pg 18, line 20: agreements -> agreement
**Reply**: Thanks. It is corrected

You may want to check that you are happy with the text size on some of your figures. It is hard to read the very small text.
**Reply**: Thank this comment. We are enlarged the font on the figures (2, 4, 5, 6, 9, 10, 11, 12).

**Response to Editor's comments**

We would like to thank you one more time for your constructive comments that has lead to improved quality and clarity of our manuscript.
You comments are repeated in black and our response is in red.
You will find the related change with track change in the new document.

pg 3, line 30: Cryostat2 -> Cryosat2
**Reply**: Thanks. We have used CryoSat-2.

pg 4, line 31 apply a post-processing -> apply post-processing
**Reply**: Thanks. It is corrected pg 8, line 8: What is superobed?
**Reply**: This is now explained (P8,L8):
"Observations are quality-controlled and superobed (i.e. the process of combining observations falling within the same model grid cell) as in Sakov et al., 2012."

pg 9, line 17: at the same
**Reply**: Thanks. It is corrected pg 9, line 18: as observations
**Reply**: Thanks. It is corrected pg 9, line 26: microwaves -> microwave
**Reply**: Thanks. It is corrected pg 9, line 30 and previous sentence: do you mean assimilation is problematic for SITs thinner than 0.4m. This is what the previous sentence suggests during the melt season. Did I miss-understand your point in the previous sentence? Perhaps reword or clarify the text here.
**Reply**: Thank you this was a mistake. The sentence is corrected (P9,L30): "and the effect of ice melting leads to saturation beyond 0.4 m (see Heygster et al. 2009)"

pg 10, line 19: consider -> used, and remove 'are used' from end of sentence
**Reply**: Thanks. This is corrected pg 10, line 29: a 100 ensemble members -> 100 ensemble members
**Reply**: Thanks. It is corrected pg 15, line 11: near the location
**Reply**: Thanks. It is corrected

JipingMac xie 27/10/2016 13:54

JipingMac xie 27/10/2016 13:54

Microsoft Office User 26/10/2016 13:54

JipingMac xie 27/10/2016 13:54

JipingMac xie 27/10/2016 13:54

Microsoft Office User 26/10/2016 14:02

JipingMac xie 27/10/2016 13:54

JipingMac xie 27/10/2016 13:54

JipingMac xie 27/10/2016 13:54

JipingMac xie 27/10/2016 13:54

Microsoft Office User 26/10/2016 14:23

JipingMac xie 27/10/2016 13:55

Microsoft Office User 26/10/2016 14:25

JipingMac xie 27/10/2016 13:55

JipingMac xie 27/10/2016 13:55

pg 16, line 14: impact of the j'th
**Reply**: Thanks. It is corrected pg 16, line 18: the March -> March
**Reply**: Thanks. It is corrected pg 16, line 21: The profiles -> Profiles
**Reply**: Thanks. It is corrected pg 16, line 22: Arctic -> the Arctic
**Reply**: Thanks. It is corrected pg 16, line 25: impacts -> impact
**Reply**: Thanks. It is corrected pg 17, line 2: Consider starting sentence 'The study in this paper' rather than 'This study'
**Reply**: Thanks. It is corrected pg 17, line 33: These studies follow from the first ...
**Reply**: Thanks. It is corrected pg 18, line 4: spun up
**Reply**: Thanks. It is corrected pg 18, line 6: at time when -> the time period
**Reply**: Thanks. It is corrected pg 18, line 20: agreements -> agreement
**Reply**: Thanks. It is corrected

You may want to check that you are happy with the text size on some of your figures. It is hard to read the very small text.
**Reply**: Thank this comment. We are enlarged the font on some of the figures.

JipingMac xie 27/10/2016 13:54

JipingMac xie 27/10/2016 13:54

JipingMac xie 27/10/2016 13:54

JipingMac xie 27/10/2016 13:54

JipingMac xie 27/10/2016 13:54

JipingMac xie 27/10/2016 13:54

JipingMac xie 27/10/2016 13:54

JipingMac xie 27/10/2016 13:54

JipingMac xie 27/10/2016 13:54

JipingMac xie 27/10/2016 13:54

JipingMac xie 27/10/2016 13:54

Microsoft Office User 26/10/2016 14:29

[revised manuscript text omitted]

Unknown

JipingMac xie 27/10/2016 09:37

JipingMac xie 27/10/2016 09:37

Unknown

[Figure]

**Fig**. 12 Same as Figure 11 for November 2014

Unknown

JipingMac xie 27/10/2016 09:37